# Modeling Dynamics over Meshes with Gauge Equivariant Nonlinear Message Passing

**Jung Yeon Park, Lawson L.S. Wong[†], Robin Walters[†]**
Khoury College of Computer Sciences
Northeastern University
Boston, MA 02115
{park.jungy@northeastern.edu, lsw@ccs.neu.edu,
r.walters@northeastern.edu}

## Abstract

Data over non-Euclidean manifolds, often discretized as surface meshes, naturally arise in computer graphics and biological and physical systems. In particular, solutions to partial differential equations (PDEs) over manifolds depend critically on the underlying geometry. While graph neural networks have been successfully applied to PDEs, they do not incorporate surface geometry and do not consider local gauge symmetries of the manifold. Alternatively, recent works on gauge equivariant convolutional and attentional architectures on meshes leverage the underlying geometry but underperform in modeling surface PDEs with complex nonlinear dynamics. To address these issues, we introduce a new gauge equivariant architecture using nonlinear message passing. Our novel architecture achieves higher performance than either convolutional or attentional networks on domains with highly complex and nonlinear dynamics. However, similar to the non-mesh case, design trade-offs favor convolutional, attentional, or message passing networks for different tasks; we investigate in which circumstances our message passing method provides the most benefit [1].

## 1 Introduction

Surfaces embedded in 3D space appear in many domains such as computer graphics [1], structural biology [2, 3], neuroscience [4], climate pattern segmentation [5, 6], and fluid dynamics [7]. Such objects are naturally Riemannian manifolds but are often discretized into meshes for computational tractability. Unlike other 3D representations such as point clouds or voxels, meshes encode both the geometry and topology of the surface.

One important task that naturally arises over meshes is solving partial differential equations (PDEs) over surfaces and non-Euclidean manifolds [11–14]. A classic approach to solving surface PDEs is to model the domain as a mesh and use the finite element method and the variational formulation [15, 16]. Recently, there has been increasing success in applying deep learning methods to accelerate solving PDEs and building differentiable surrogates for dynamics models [17–20]. While there are many effective approaches for gridded data or point clouds, fewer methods exist for leveraging the geometric structure of meshes. Several approaches simply treat the mesh as a graph by approximating patches of the manifold as Euclidean [21–23]. However, these approaches do not exploit the geometric properties of meshes, and Verma et al. [24], de Haan et al. [9] have shown that not retaining this information leads to poor performance. One way to express such information is to specify the local

---

[†]Equal advising.
[1]Project website with code: https://jypark0.github.io/hermes

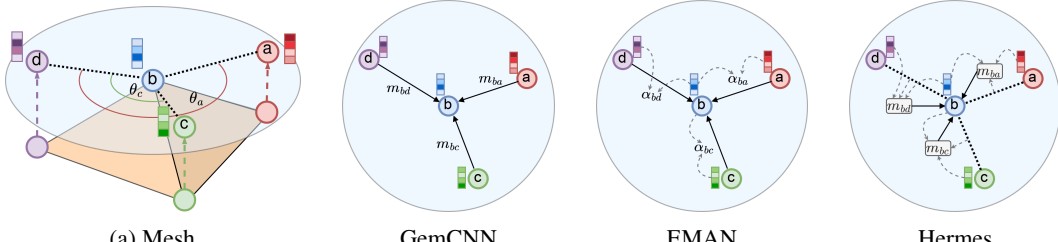

| (a) Mesh | GemCNN | EMAN | Hermes |

Figure 1: Mesh example. The neighbors of vertex $b$ are projected onto the tangent plane to compute their local orientation $(\theta_a, \theta_c)$ after choosing a reference neighbor (gauge). There are 3 flavors of gauge equivariant message passing, cf. Fig. 17 in Bronstein et al. [8]. GemCNN [9] computes linear convolutions using the neighbor node features, EMAN [10] uses linear messages with attention, while Hermes performs nonlinear message passing. Self-interactions are omitted for clarity.

gauge symmetries and explicitly model operations that are equivariant to these transformations [6, 9], by using parallel transport and applying a gauge equivariant filter.

Although previous work on gauge equivariance for meshes including convolutional (GemCNN [9]) and attentional (EMAN [10]) methods have shown good performance on shape correspondence and mesh classification tasks, we find they are inadequate for modeling complex dynamics on meshes, such as solving surface PDEs, which can be highly nonlinear. In this work, we introduce a new architecture, Hermes, that combines gauge equivariance with nonlinear message passing. As Hermes can express linear convolutions or attention, Hermes is strictly more expressive than GemCNN and EMAN. Hermes completes the 3 flavors of gauge equivariant message passing, analogous to graph neural networks [8]. See Figure 1 for a comparison.

We evaluate Hermes on several linear and nonlinear partial differential equations, and also on shape correspondence, object interaction system, and cloth dynamics. Our experiments show that Hermes outperforms convolutional or attentional counterparts in most domains, particularly on nonlinear surface PDEs. We find that Hermes is more robust to both the fineness of the mesh and surface roughness compared to both GemCNN and EMAN. We further investigate when the added model complexity of nonlinear message passing outweighs the cost. With this work, we hope to help guide practitioners on when to use nonlinear message passing schemes over linear schemes.

Our contributions are summarized as follows: 1) we propose a novel gauge equivariant, nonlinear message passing architecture, Hermes, for learning on meshes, 2) when evaluated on complex and nonlinear dynamics such as surface PDEs, Hermes outperforms both convolutional and attentional architectures and can generate realistic prediction rollouts, and 3) we investigate in which situations nonlinear message passing should be preferred over convolutional or attentional counterparts.

## 2 Background

In this work, we focus on learning signals over meshes. Similar to de Haan et al. [9], our goal is to define a model in coordinates that are intrinsic to the 2D mesh instead of using the extrinsic 3D coordinates of the embedding space. This avoids dependence on the 3D embedding, which is irrelevant. However, in order to encode data over the mesh, it is still necessary to make a choice of local coordinate frame at each vertex. Since this coordinate frame is arbitrary, it follows that the model should be invariant to change of coordinates frame at each vertex, i.e. gauge equivariant.

### 2.1 Data over Meshes

**The Mesh Datum** A mesh $M$ consists of $(\mathcal{V}, \mathcal{E}, \mathcal{F})$, where $\mathcal{V}$ is the set of vertices, $\mathcal{E} = \{(i, j)\}$ is the set of ordered vertex indices $i, j$ connected by an edge, and $\mathcal{F} = \{(i, j, k)\}$ is the set of ordered vertex indices $i, j, k$ connected by a triangular face. Let $\mathcal{N}_v$ denote the set of vertices connected to a vertex $v$. We assume that the mesh represents a discrete 2-dimensional manifold embedded in $\mathbb{R}^3$ (i.e. a manifold mesh). Let $x_b \in \mathbb{R}^3$ be the coordinates of vertex $b$. We assign a normal vector $n_b$ to each vertex $b$ equal to the normalized average of the surface normals of the faces that contain the vertex. The associated tangent plane $T_b M$ at vertex $b$ is the 2-dimensional affine space through $x_b$ orthogonal to the normal $n_b$.

**Local Coordinate Frame**    We adopt the strategy of [9] by defining the local coordinate frame at each vertex in terms of a reference neighboring vertex $d \in \mathcal{N}_b$. See Figure 1a for a visualization. The reference neighbor defines a positively oriented orthonormal basis $\{e_1^b, e_2^b\}$ for the tangent space $T_b M$. The orientation of every neighbor $v \in \mathcal{N}_b$ can be represented with polar angles, where $\theta_v$ is the angle between $v$ and the reference neighbor after projection to the tangent plane. In Figure 1a, $\theta_d = 0$ as $d$ is itself the reference neighbor and we show the $\theta_v$ for the other neighbors with respect to the reference neighbor $d$. See Appendix A.1 for more information.

**Change of Gauge**    A different choice of reference neighbor will result in a different positively oriented orthonormal basis. Any two such bases will be related by an element $g \in \mathrm{SO}(2)$. Here $\mathrm{SO}(2)$ is called the gauge group and $g$ is the gauge transformation. Note that such a change is local, and a different change $g^p$ can be performed at each vertex $p \in \mathcal{V}$. For example, if we switch from reference neighbor $d$ to $c$, then we induce a gauge transformation $\mathrm{Rot}(\phi) \in \mathrm{SO}(2)$ where $\phi = \theta_c - \theta_d$ is the angle between $d$ and $c$. The orientations are then updated $\theta_q \mapsto \theta_q - \phi, \forall q \in \mathcal{N}_p$.

**Feature Fields on the Mesh**    Input, output, and hidden features are encoded as data over the vertices $\{f_p\}_{p \in \mathcal{V}}$ or edges of the mesh $\{e_{pq}\}_{(p,q) \in \mathcal{E}}$. We consider directed graphs and so for the edge feature $e_{pq}$, $p$ is the source and $q$ is the target node. We situate $e_{pq}$ to be at $T_p M$ for simplicity as they are often chosen to be vectors relative to $p$ or the edge distance [25, 18]. A scalar field, such as temperature, is invariant to change of gauge, and thus $f_p \in \mathbb{R}$ transforms according to the trivial representation $\rho_0$ of $\mathrm{SO}(2)$. A vector field over the mesh, representing e.g. a flow across the manifold would have $f_p \in \mathbb{R}^2$ and transform according to the standard rotation representation $\rho_1$ of $\mathrm{SO}(2)$ by $2 \times 2$ matrices. In general, feature fields may be any representation $\rho_n$ of the gauge group $\mathrm{SO}(2)$.

**Parallel Transport**    Let $f_p$ and $f_q$ be feature vectors at nodes $p, q$, respectively, where $q$ is adjacent to $p$. As features at different nodes live in different tangent spaces and have different bases, in order for the anisotropic kernel to be applied to $f_q$ for each $q \in \mathcal{N}_p$, each $f_q$ must be parallel transported to $T_p M$ and be in the same gauge. The parallel transporter $g_{q \to p}$ first aligns the tangent plane at $q$ to the tangent plane at $p$ by a 3D rotation and then transforms the gauge at $q$ to the gauge at $p$ by a planar rotation. Acting on $f_q$ with $g_{q \to p}$ transports $f_q$ to the gauge at $p$. For the edge features $e_{pq}$, no parallel transport is required as the features live in $T_p M$. For more details, see Appendix A.2 and Appendix A.2 of [9].

## 2.2   Flavors of Message Passing

Meshes are graphs with additional geometric information. Neural networks designed for meshes may be understood by analogy to graph neural networks (GNNs). As outlined in Bronstein et al. [8], there are largely three flavors of GNNs: convolutional, attentional, and nonlinear message-passing.

The three images on the right of Figure 1 show the three different GNN flavors in the context of gauge equivariant networks over meshes. To date, both convolutional and attentional GNNs have gauge equivariant analogues for learning over meshes. Here, we introduce a gauge equivariant mesh network in the nonlinear message-passing flavor. Let $p$ be the source node ($b$ in Figure 1a), $q \in \mathcal{N}_p$ be the target nodes (one-hop neighborhood), and $f$ the signals over nodes.

**Convolutional**    GemCNN [9] uses convolutions where the kernel $K_{\mathrm{neigh}}$ is applied (anisotropically) to features at the target nodes. It is comparable to graph convolutional networks [26] but with anisotropy and gauge equivariance. As convolutions are linear, the messages passed to the source node are linear, and these messages are aggregated in a permutation invariant way. GemCNN also has a kernel $K_{\mathrm{self}}$ to model self-interactions before updating the feature at the source node. The gauge equivariant convolution (GemCNN layer) is defined as

$$f_p' = K_{\mathrm{self}} f_p + \sum_{q \in \mathcal{N}_p} K_{\mathrm{neigh}}(\theta_q) \rho_{\mathrm{in}}(g_{q \to p}) f_q, \tag{1}$$

where $K_{\mathrm{neigh}}(\theta_q)$ is the kernel for $q$ and $\rho_{\mathrm{in}}(g_{q \to p}) f_q$ is the feature vector at $q$ parallel transported to the gauge at $p$. Input features $f_q$ transform according to $\rho_{\mathrm{in}}$ and output features $f_q'$ according to $\rho_{\mathrm{out}}$.

The filter $K_{\mathrm{neigh}}$ is anisotropic, i.e., it depends on the orientation $\theta_q$. This is strictly more expressive than an isotropic filter (as in a GNN) which would simply use the same weights for each neighboring vertex. However, the dependence on orientation means that without constraints, the convolution depends on the choice of local coordinate frame. To fix this and impose gauge equivariance, $K_{\mathrm{neigh}}$ must satisfy $K = \rho_{\mathrm{out}}(g)^{-1} K \rho_{\mathrm{in}}(g)$. We also define the kernel to only depend on the orientation and

not on the radial distance of neighboring nodes, as including the radius in the parameterization was not beneficial [9] and verified in our experiments.

**Attentional**   EMAN [10] extends GemCNN by adding an attention mechanism to the messages before the aggregation step, analogous to graph attention networks [27]. Gauge equivariant attention with the self-interaction term is defined as

$$f'_p = \alpha_{pp} K^{\text{self}}_{\text{value}} f_p + \sum_{q \in \mathcal{N}_p} \alpha_{pq} K^{\text{neigh}}_{\text{value}}(\theta_q) \rho(g_{q \to p}) f_q, \tag{2}$$

where $\alpha_{pp}, \alpha_{pq}$ are the attention weights for the self-interaction and neighbor interaction terms (see Equation (12) in Appendix B.2). Multi-headed attention can be used by concatenating the results of the individual heads. The attention weights are combined linearly with the neighboring node features and can be seen as weighted local averaging. Note that due to the separate kernels for the keys, queries, and values, an EMAN layer uses many more parameters than an equivalent GemCNN layer with the same dimensions.

## 3   Gauge Equivariant Nonlinear Message Passing

We propose gauge equivariant nonlinear message passing for meshes, complementing the convolutional and attentional gauge equivariant methods. As is the case for nonlinear message passing GNNs, our network is designed to be better suited for tasks with complex local interactions. Our network, named Hermes, consists of a sequence of message passing blocks, each of which contains an edge network $\phi_e$, an aggregation (we use sum throughout), and a node network $\phi_n$.

The edge network models neighboring interactions and takes as inputs the source node features $f_p$, target node features $f_q$ where $q \in \mathcal{N}_p$, and edge features $e_{pq}$ where $(p, q) \in \mathcal{E}$. The target node features are parallel transported to the gauge at $p$ as $\rho(g_{q \to p})$. The network $\phi_e$ consists of $N_e$ gauge equivariant convolutions followed by regular nonlinearities [9] which also preserve gauge equivariance. The nonlinear messages $m_{pq}$ are then aggregated to $m_p$. The node network models self-interactions and takes as input $m_p \oplus f_p$, where $\oplus$ represents the direct sum, and then applies $N_n$ gauge equivariant convolutions with $K_{\text{self}}$ kernels and regular nonlinearities. Equations (3)-(7) define gauge equivariant nonlinear message passing,

$$h_{pq} = f_p \oplus \rho(g_{q \to p}) f_q \oplus e_{pq}, \qquad\qquad \forall (p, q) \in \mathcal{E} \tag{3}$$

$$m_{pq} = \phi_e(h_{pq}) = \sigma \circ K^{N_e}_{\text{neigh}}(\theta_q) \circ \cdots \circ \sigma \circ K^1_{\text{neigh}}(\theta_q)(h_{pq}), \tag{4}$$

$$m_p = \sum_{q \in \mathcal{N}p} m_{pq}, \qquad\qquad \forall p \in \mathcal{V} \tag{5}$$

$$h_p = m_p \oplus f_p \tag{6}$$

$$f'_p = \phi_n(h_p) = \sigma \circ K^{N_n}_{\text{self}} \circ \cdots \circ \sigma \circ K^1_{\text{self}}(h_p), \tag{7}$$

where $h_{pq}$ and $h_p$ are inputs to $\phi_e$ and $\phi_n$, $\sigma$ is the regular nonlinearity, and $\circ$ denotes composition.

Figure 2 shows the Hermes architecture for one of the datasets used in experiments (wave PDE, see Section 4.1). For comparison, we also include the GemCNN and EMAN architectures in Appendix B. A residual connection is included at the end of each HermesBlock for more expressivity, see Section 5.5 for ablation results.

**Proof of Gauge Equivariance**   We explicitly show that Hermes is equivariant to local gauge transformations in Proposition 1.

**Proposition 1.** *Hermes is equivariant to local gauge transformations $g^p$ for any $p \in \mathcal{V}$ of a mesh.*

The proof, provided in Appendix B.3, uses the fact that all of the kernels used in Hermes satisfy the constraints for gauge equivariance. The nonlinearities are exactly gauge equivariant as the number of intermediate samples $N$ used in the discrete Fourier transform approaches infinity and approximately gauge equivariant for finite $N$ (see Section 4 in [9] for more details). Thus Hermes is also gauge equivariant (in the limit).

Hermes combines both nonlinear message passing and gauge equivariance, generalizing GemCNN and EMAN, and completes the full picture with the 3 flavors of message passing (cf. Figure 1). An

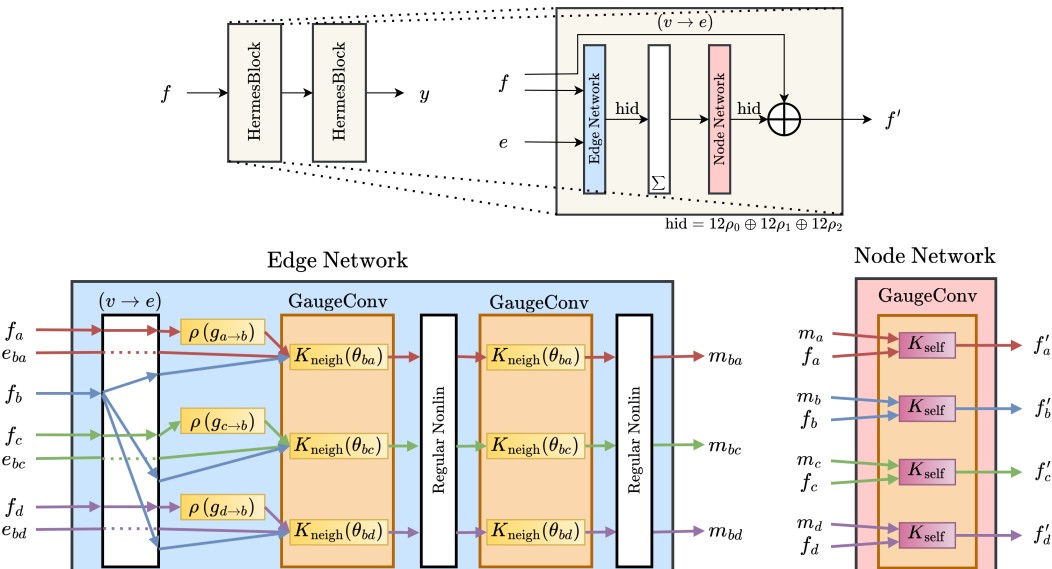

Figure 2: Hermes network architecture for the Wave PDE dataset. There are two message blocks, each with 2 layers for the edge network $\phi_e$ and 1 layer for the node network $\phi_n$. To illustrate the computations within the edge and node networks, we use the example mesh from Figure 1a as input. In the edge network, we show only the computations for node $b$ for clarity.

important point to note is that nonlinear message passing decouples the effect of the number of layers in the network (depth) with the receptive field of the graph. For graph convolutions using 1-hop neighbors, the receptive field of messages is exactly equal to the network depth. In contrast, nonlinear message passing computes messages using features from neighbors at an arbitrary hop distance away from the source node before the message is passed to the source node. We hypothesize that this decoupling is beneficial for tasks with complicated interaction dynamics between node neighbors, and previous works with graph message passing networks [28, 29] have shown good performance for abstract graphs (non-spatial graphs) involving objects. Although meshes are inherently spatial, we show that nonlinear message passing is an important tool for solving PDEs on surfaces and can outperform either convolutional or attentional mechanisms.

## 4 Experiment Design

To evaluate gauge equivariant nonlinear message passing, we consider several different tasks. Our primary domains are partial differential equations on meshes, but we also consider shape correspondence, object interactions, and cloth simulation. Throughout, we consider triangular meshes as they are the most common. Further details and visualizations of the domains are provided in Appendix C.

### 4.1 Domains

**Partial Differential Equations on meshes** We consider three linear and nonlinear surface partial differential equations (PDEs), where the dynamics occur on the surface of an object, represented as a two dimensional mesh embedded in 3D space. Since solving PDEs on meshes, e.g. heat diffusing over a surface, naturally depends on the intrinsic mesh geometry, gauge-equivariant nonlinear message passing is a promising solution. For all three equations, we use example meshes in the PyVista library [30] and generate 5 trajectories with different initial conditions, see Appendix C for more details.

Heat/Wave equation: The heat and wave equations are second-order linear partial differential equations (see Appendix C.1). Solving the heat equation on a surface mesh can correspond to modeling the dynamics when an external hot object makes contact at certain points on a thin hollow object. The wave equation describes how acoustic waves propagate on a thin hollow surface. The dynamics of both equations are highly dependent on the local mesh geometry.

Cahn-Hilliard equation: The Cahn-Hilliard equation [31] describes phase separation in a binary fluid mixture and is often used to model spinodal decomposition. It is a fourth-order, nonlinear,

time-dependent PDE and is often factored into two coupled second-order equations. The surface Cahn-Hilliard equation can model real-world applications such as cell proliferation [32] and two-component vesicles [33]. As a nonlinear PDE, we expect our nonlinear message passing method to exhibit greater performance than other linear message passing flavors.

**Shape correspondence**   As a standard mesh benchmark, we use the same FAUST dataset used in previous work [9, 10]. The dataset consists of 80 train and 20 test high resolution scans of 10 humans in 10 different poses. The task is to determine shape correspondence between different meshes. As the vertices are all registered and represent the same position on the human body, this task is equivalent to classifying the correct label for each vertex.

**Object interactions**   Inspired by interaction systems [28, 29, 34], we consider complex dynamics of interacting objects on a mesh. On a coarse triangular mesh with random hills, each object occupies a vertex and is oriented towards a neighboring vertex. An action can either turn the object left (changing its orientation), move the object forward, or turn right. Objects cannot move forward if there is another object at the destination vertex, giving rise to complex interacting dynamics. Furthermore, we consider the geometry of the mesh such that an object cannot move forward if the height difference between the current vertex and destination vertex is too high, or if the angle between the vertex normals is too large. If an object is able to move forward, we parallel transport its orientation and then choose the nearest neighboring node as its new orientation.

**FlagSimple**   We also include the `FlagSimple` dataset from [20], which simulates cloth dynamics of a flag with self-collisions. Unlike the other datasets, the mesh is dynamic where the node positions change over time. The dataset was created using ArcSim [35] with regular meshing over 400 timesteps. See Appendix A.1 of [20] for more information.

### 4.2   Training Details

For the PDE datasets, we report test root mean squared error (RMSE) of the prediction at the next timestep given the previous 5 timesteps. We use three separate test datasets and evaluate generalization to future timesteps (`test time`), unseen initial conditions (`test init`), and unseen meshes (`test mesh`). For `test time`, we train on timesteps $T = 0, \ldots, 149$ and test on $T = 150, \ldots, 200$. For `test init`, we test on trajectories with new initial conditions. For `test mesh`, we evaluate on completely unseen meshes to see whether a method overfits to specific mesh geometries.

For baselines, our main comparison is with GemCNN and EMAN to gauge how beneficial nonlinear message passing is over convolutional or attentional flavors. We also compare against 1) a SOTA messsage passing, mesh-aware method MeshGraphNet [20], 2) an E(3)-equivariant, non mesh-aware message passing network EGNN [25], 3) a non-equivariant, mesh-aware baseline SpiralNet++ [36], and 4) standard non-equivariant GNNs: graph convolutional networks (GCN) [26] and message passing networks (MPNN) [37, 38]. We tune each architecture to use a similar number of trainable parameters for a fair comparison, see Table 9 in Appendix D. A more detailed feature comparison of each method is provided in Table 8 and all other training details are relegated to Appendix D.

## 5   Results

Table 1 shows the results on the PDE datasets and other results are given in Table 10 in Appendix E. On Heat, we see that EMAN outperforms GemCNN and Hermes outperforms EMAN significantly. This coincides with our expectation as attention mechanisms can express convolutions using constant weights, and Hermes generalizes both linear convolution and attention mechanisms. However, this does not hold for the Wave dataset, where EMAN is noticeably worse than GemCNN while Hermes still performs well. On the nonlinear Cahn-Hilliard dataset, we see that EMAN cannot generalize well to unseen meshes. Overall Hermes achieves an RMSE approximately 3 times lower than that of GemCNN, and between 2 to 8 times lower than that of EMAN.

Compared to other non gauge equivariant baselines, Hermes outperforms all methods, except for certain cases with MeshGraphNet. Hermes is worse than MeshGraphNet on Heat, outperforms on Wave, and performs similarly on Cahn-Hilliard. It performs substantially better on all test mesh datasets, which may indicate that Hermes can generalize to the true dynamics function, rather than the specific dynamics seen in the training trajectories. In other words, Hermes is better adapted to use the underlying geometry while MeshGraphNet overfits to specific geometries.

Table 1: RMSE on PDE domains, using 5 runs. All values are expressed in $\times 10^{-3}$ and gray denotes 95% confidence intervals. Hermes generally outperforms baselines, except for some cases with MeshGraphNet (MGN). Hermes outperforms MeshGraphNet on all test mesh datasets.

|  |  | **Hermes** | GemCNN | EMAN | GCN | MPNN | MGN | EGNN | SpiralNet++ |
|---|---|---|---|---|---|---|---|---|---|
| HEAT | Test time | $1.18_{\pm 0.3}$ | $3.88_{\pm 0.8}$ | $2.93_{\pm 0.9}$ | $152_{\pm 1.2}$ | $2.66_{\pm 0.8}$ | $\mathbf{0.93}_{\pm 0.2}$ | $3.09_{\pm 1.2}$ | $2.82_{\pm 0.2}$ |
|  | Test init | $1.16_{\pm 0.3}$ | $3.85_{\pm 0.8}$ | $2.90_{\pm 0.9}$ | $152_{\pm 0.9}$ | $2.63_{\pm 0.8}$ | $\mathbf{0.93}_{\pm 0.2}$ | $3.07_{\pm 1.2}$ | $6.44_{\pm 0.1}$ |
|  | Test mesh | $\mathbf{1.01}_{\pm 0.3}$ | $3.50_{\pm 0.6}$ | $2.47_{\pm 0.7}$ | $127_{\pm 2.2}$ | $2.36_{\pm 0.7}$ | $2.41_{\pm 1.1}$ | $8.96_{\pm 5.0}$ | $22.0_{\pm 0.3}$ |
| WAVE | Test time | $\mathbf{5.43}_{\pm 0.8}$ | $12.2_{\pm 1.5}$ | $19.0_{\pm 3.0}$ | $162_{\pm 5.0}$ | $9.07_{\pm 1.2}$ | $6.26_{\pm 0.9}$ | $45.9_{\pm 6.1}$ | $8.88_{\pm 1.2}$ |
|  | Test init | $\mathbf{3.72}_{\pm 1.3}$ | $7.28_{\pm 0.7}$ | $15.3_{\pm 3.6}$ | $158_{\pm 5.9}$ | $5.24_{\pm 1.1}$ | $4.24_{\pm 0.6}$ | $12.1_{\pm 3.5}$ | $8.47_{\pm 0.6}$ |
|  | Test mesh | $\mathbf{3.79}_{\pm 1.3}$ | $8.23_{\pm 0.6}$ | $15.8_{\pm 3.7}$ | $164_{\pm 5.1}$ | $6.29_{\pm 1.3}$ | $7.01_{\pm 1.9}$ | $54.5_{\pm 18}$ | $10.8_{\pm 0.8}$ |
| CAHN-HILLIARD | Test time | $\mathbf{3.48}_{\pm 0.9}$ | $13.8_{\pm 12}$ | $8.41_{\pm 4.0}$ | $250_{\pm 7.6}$ | $7.25_{\pm 3.1}$ | $\mathbf{4.49}_{\pm 0.6}$ | $8.36_{\pm 1.4}$ | $11.6_{\pm 3.3}$ |
|  | Test init | $4.23_{\pm 1.5}$ | $14.9_{\pm 12}$ | $8.75_{\pm 4.0}$ | $383_{\pm 6.0}$ | $7.52_{\pm 3.0}$ | $\mathbf{4.64}_{\pm 0.6}$ | $10.6_{\pm 1.1}$ | $12.9_{\pm 2.8}$ |
|  | Test mesh | $\mathbf{5.41}_{\pm 0.8}$ | $14.1_{\pm 12}$ | $43.8_{\pm 27}$ | $391_{\pm 8.6}$ | $7.63_{\pm 3.0}$ | $18.7_{\pm 7.2}$ | $9.38_{\pm 1.7}$ | $13.4_{\pm 2.5}$ |

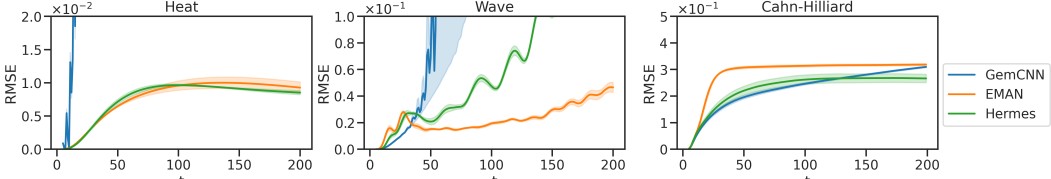

Figure 3: Errors from long-horizon prediction rollouts on unseen meshes, given only initial conditions. Error bars denote standard error over 5 runs averaged over unseen meshes (2 meshes for heat/wave, 1 for Cahn-Hilliard). GemCNN has exploding errors with increasing $t$ on Heat and Wave. Hermes generally outperforms GemCNN and EMAN, with the exception of EMAN on Wave.

On the FlagSimple dataset (Table 10), Hermes outperforms MeshGraphNet considerably, suggesting that Hermes is not limited to static meshes and can handle temporally changing meshes well. Note that MeshGraphNet was modified to have a similar number of parameters as Hermes and so the results are different than reported in Pfaff et al. [20].

## 5.1 Long-horizon prediction rollouts

Using a representative random run, we generate predictions from each model autoregressively given only the initial conditions. We generate predictions on the unseen meshes (test mesh) and roll out the entire trajectory of the PDE ($T_{\max} = 200$). Figure 3 shows how the errors change with the increasing rollout horizon. For GemCNN, the errors accumulate quickly and the predictions diverge for Heat and Wave. EMAN performs similarly to Hermes on Heat but underperforms on Cahn-Hilliard. All methods eventually diverge on Wave; this may be due to the fact that the wave amplitude oscillates multiple times over the longer horizon and so it may be more difficult to predict the periodic nature.

Table 2 shows prediction samples at $T + 50$ generated autoregressively given only initial conditions. GemCNN fails on the Heat dataset and does not produce the correct spatial patterns for Wave and Cahn-Hilliard. EMAN performs well on Wave, but fails on Cahn-Hilliard. Hermes gives fairly realistic predictions on all datasets. See Appendix E.1 for more samples with varying $T$.

## 5.2 Mesh Fineness

Here we investigate how mesh fineness impacts performance with respect to the three flavors of message passing. Our intuition is that as meshes become finer, the features at each node become more similar. Thus the dynamics between nodes would become more linear and convolutional approaches may perform comparably with nonlinear message passing.

For this experiment, we solve the heat and wave equations on a single mesh ("armadillo"). To generate different mesh resolutions, we simplify the original mesh (# vertices = $172{,}974$) to $\{348, 867, 1731, 3461, 8650\}$ vertices using quadric decimation [39] (1731 vertices were used for the main results in Table 1). See Figure 9 in Appendix E.2 for data visualizations. We generate 15 trajectories with $T_{\max} = 100$ and use 5 of the 15 trajectories as `test init` and use $T = 81, \ldots, 100$ of the remaining 10 trajectories for the `test time`. As we use a single mesh, we do not test generalization to unseen meshes. We use 3 random seeds for each method.

Table 2: Qualitative prediction samples rolled out to the full path using only initial conditions. GemCNN completely fails on Heat, while EMAN fails on Wave. Hermes predicts the spatial patterns accurately and outperforms GemCNN and EMAN on all datasets.

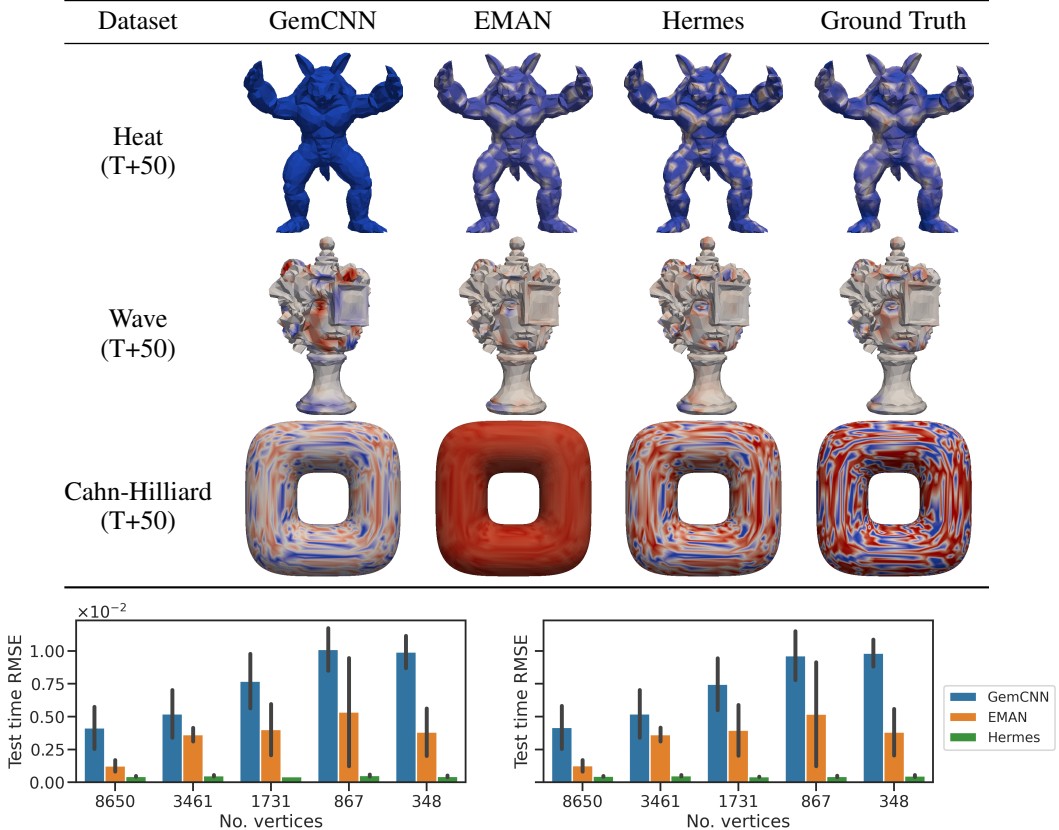

Figure 4: Performance for varying mesh fineness for heat on the test time (left) and test init (right) datasets. Error bars denote standard error over 3 runs. The errors for GemCNN and EMAN increase as the mesh becomes coarser, while Hermes performs similarly throughout.

Figure 4 shows the results for the heat dataset. RMSE on both the test time and test init datasets increase for GemCNN and EMAN as the mesh becomes coarser (decreasing number of vertices). This coincides with the intuition that coarser meshes have more nonlinear dynamics between nodes and thus contributing to the increase in errors. On the other hand, Hermes still quantitatively outperform GemCNN and EMAN across all mesh resolutions. It is also robust to varying mesh fineness and has similar error values throughout.

On the wave dataset, we find that Hermes still outperforms GemCNN and EMAN at each resolution though the gap is not as large, see Figure 10 (Appendix E.2). There is a surprising decreasing trend in errors for all methods as the mesh becomes finer. This indicates that mesh fineness may not be the only factor at play: it is possible that the type of dynamics (PDE) used and the specific architecture (e.g. graph receptive field) can affect the results. Another possible explanation is that as we use a single simplified mesh, there may be some artifacts that affect the wave PDE simulations.

## 5.3 Surface Roughness

We investigate whether surface roughness affects the performance gap between GemCNN and Hermes. We use the same armadillo mesh with $1,731$ vertices from before and extract the vertex normals. The vertex normals are then randomly scaled using a Gaussian distribution $\mathcal{N}(0, s^2)$, where $s \in \{0.1, 0.5, 1, 1.5, 3\}$. The vertex coordinates are then modified by adding these scaled normals to the original coordinates, resulting in different surface roughnesses. See Figure 11 in Appendix E.3 for visualizations. We use the same settings as in the fineness experiments in Section 5.2.

Similarly to the results on mesh fineness, we find that Hermes retains the performance advantage over GemCNN and EMAN across all roughness scales. Its RMSE errors are also nearly constant

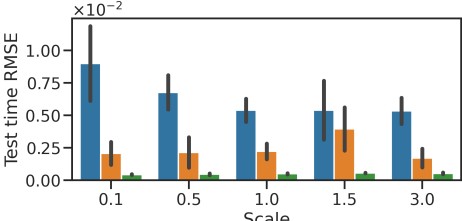 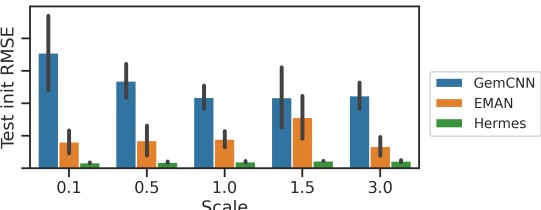

Figure 5: Performance for varying surface roughness for heat on the test time (left) and test init (right) datasets. Error bars denote standard error over 3 runs. Increasing scale increases the surface roughness of the mesh. Hermes outperforms GemCNN and EMAN across different roughness scales and is much more robust.

Table 3: Mean forward computation time in seconds (ms) during inference on test time PDE datasets.

|  | GemCNN | EMAN | Hermes | GCN | MPNN | MeshGraphNet | EGNN | SpiralNet++ |
|---|---|---|---|---|---|---|---|---|
| HEAT (ms) | 12.4 | 19.5 | 10.8 | 1.5 | 1.2 | 2.2 | 1.4 | 0.9 |
| WAVE (ms) | 12.5 | 17.7 | 10.5 | 1.4 | 1.3 | 2.2 | 1.3 | 0.9 |
| CAHN-HILLIARD (ms) | 6.9 | 9.2 | 6.2 | 1.9 | 1.5 | 2.1 | 1.6 | 1.3 |

throughout indicating robustness to surface roughness. Surprisingly, GemCNN seems to perform increasingly better as the surface becomes rougher while EMAN seems to perform roughly equal. On the wave dataset, Hermes still outperforms other baselines at most roughness scales, see Figure 12 in Appendix E.3. There does not seem to be any correlation in errors with surface roughness.

## 5.4 Computation Time

As the edge and node networks $\phi_n, \phi_e$ consist of multi-layer perceptrons, one might think Hermes requires more computation time than its simpler convolutional or attentional counterparts. Table 3 shows that this is not the case; Hermes has a lower computation time than both GemCNN and EMAN. As we control for a similar number of parameters in each architecture, there are fewer aggregations in Hermes than in GemCNN or EMAN resulting in a lower average computation time in the forward pass. For EMAN in particular, due to its attention mechanism, we find that it uses many more parameters than either GemCNN or Hermes. Thus constraining EMAN to have a similar number of parameters as GemCNN restricts its expressivity. On the other hand, Hermes is very flexible as it can use a different number of layers for the edge and node networks. Even though Hermes uses a slightly lower hidden dimension than GemCNN in all domains, the results show that this does not hamper model expressivity and it still achieves higher performance. As expected, all three gauge equivariant methods are significantly more computationally expensive than standard graph neural networks.

## 5.5 Residual connection ablation

As Hermes models the self-influence of nodes, we also test whether a residual connection is necessary. Table 12 in Appendix E.4 shows the ablation of the residual connection. Having a residual connection improves performance on Heat, but decreases performance slightly on Wave and Cahn-Hilliard. Thus the residual connection seems to be task-dependent and should be considered a hyperparameter. In our experiments, a residual connection was used for each message passing block.

## 6 Related Work

**Learning over meshes**    Several different mesh operators have been studied within computer graphics, in the context of shape classification [40–42], dense shape correspondence [43, 36], mesh segmentation [44–46]. Due to the success of graph neural networks (GNNs) [26, 27, 38], several approaches use GNNs to process meshes [21, 22, 47, 36] and incorporate geometric information via geodesic convolutions, anisotropic kernels, dual graphs, or spiral operators. Pfaff et al. [20] introduce a state of the art method for learning simulations with meshes by representing a graph in mesh space and in world space. In this work, we introduce a novel architecture that combines nonlinear message passing with explicit equivariance to local gauge transformations.

**Gauge Symmetry**    Most works on non-Euclidean manifolds have used convolutions on discretized patches of the manifold, approximating them to be Euclidean. Masci et al. [21] define the patch operator using local geodesic coordinate systems. Monti et al. [23] use a mixture of parametric Gaussian kernels. All of these works use (linear) convolutions, unlike our work. Boscaini et al. [22] uses anisotropic diffusion kernels, but uses the principal curvature direction as the preferred gauge which may be ill-defined for some shapes. On the other hand, several works have considered equivariance to gauge symmetry. Cohen et al. [6] proposes gauge equivariant convolutions on the icosahedron. Most similar to our work is that of [9, 48, 10], which explicitly incorporate gauge equivariant kernels for meshes. de Haan et al. [9] use convolutions while He et al. [48] and Basu et al. [10] use attention to discrete and continuous gauge transformations, respectively. In this work, we complete the picture and propose general nonlinear message passing with gauge equivariance for meshes. For more in-depth theory and discussion of local gauge equivariance on manifolds, see [49].

**Solving PDEs on surfaces**    With the rise of physics-informed neural networks [50–52] and their increased sample efficiency and performance over classical methods, several recent works have focused on solving complex partial differential equations with deep learning, e.g. fluid dynamics [53–55], thermodynamics [56], structural mechanics [57, 58], and material science [59, 60]. Some approaches learn the finite-dimensional [51, 53, 54] or infinite-dimensional [61, 17, 62] solution operators. However, there has been less work on applying deep learning to solving PDEs on surfaces embedded in 3D space using meshes, with some exceptions [63–65]. Fang and Zhan [63] solve the Laplace-Beltrami equation over a 3D surface with neural networks while Tang et al. [64] propose an extrinsic approach. However, both methods are mesh-free and only use points and their normals, discarding any connectivity information. Li et al. [65] extend Fourier neural operator [66] and learns a diffeomorphic deformation between the input space and the computational mesh. While Geo-FNO depends on the embedding space of the mesh (e.g. embedding a rough 2D mesh in 3D), our method works directly on the intrinsic mesh surface. Perhaps most relevant to this paper is [67], which extends GemCNN [9] in several ways to predict hemodynamics on artery walls. In this work, we propose a nonlinear message passing architecture for meshes that retains the underlying geometry and demonstrate their effectiveness in predicting a variety of surface dynamics.

# 7    Discussion

We introduce a novel architecture, Hermes, that performs gauge equivariant, nonlinear message passing for meshes. Hermes complements convolutional GemCNN and attentional EMAN and completes development of the 3 flavors of message passing in the gauge equivariant setting. In the context of meshes, similar to GNNs, there seems to be a tradeoff between simple linear operations such as convolutions versus nonlinear message passing. Convolutions are computationally efficient and can perform well on simpler tasks such as shape correspondence, see Table 10 in Appendix E. However, when the interactions are more complicated such as in surface PDEs, nonlinear message passing surpass linear schemes. By decoupling the degree of nonlinearity from the depth of the network and receptive field, Hermes outperforms GemCNN and EMAN significantly on the PDE datasets and produces realistic predictions given only initial conditions.

A limitation is that Hermes may use more parameters depending on the architectures of the edge and node networks. In particular, both EMAN and Hermes cannot scale well to meshes with a large number of vertices naively, and one may need to consider more sophisticated approaches such as multi-scale or graph expander approaches. Furthermore, the architecture search space for Hermes is larger than that of GemCNN or EMAN, as one needs to consider different combinations of numbers of layers in the edge and node networks, along with the number of message passing blocks.

For future work, one direction is to consider different dynamics such as non-stationary or chaotic dynamics, and other PDEs important in real world applications such as Navier-Stokes for climate or blood flow. Another direction is to analyze the design space of gauge equivariant networks. While GNNs have been extensively studied, far less work exists for mesh methods. GNNs are often highly task-specific and there are many design dimensions (e.g. residual connections, message passing iterations, etc.) to consider [68]. It would be particularly helpful for practitioners to have guidelines on when to use gauge equivariance or message passing over simpler approaches. This work aims to be a first step in this direction by demonstrating Hermes as a good fit for predicting nonlinear dynamics on meshes.

## Acknowledgments and Disclosure of Funding

We sincerely thank Vedanshi Shah for the initial data generatìon code and Pim de Haan for clarifying parts of GemCNN. This work is supported in part by NSF 2107256 and 2134178. This work was completed in part using the Discovery cluster, supported by Northeastern University's Research Computing team.

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

## A  Background

### A.1  Local Coordinate Frame

As discussed in the main text, the reference neighbor $d$ defines the basis $\{e_1^b, e_2^b\}$ for the tangent space $T_b M$ (see Figure 1a). Denote by $\log_b(x_d) = \mathrm{Proj}_{T_b M}(x_d) - x_b$ the vector from $x_b$ (the coordinates of $b$) to the orthogonal projection of $x_d \in \mathbb{R}^3$ (the coordinates of $d$) onto the tangent plane $T_b M$. Let $e_1^b = \log_b(x_d)/\|\log_b(x_d)\|$ and $e_2^b = n_b \times e_1^b$. Then the orientation of every neighbor $v \in \mathcal{N}_b$ can be represented with polar angles, where

$$\theta_v = \mathrm{atan2}\Big(e_2^{b\top} \log_b(v), e_1^{b\top} \log_b(v)\Big) \tag{8}$$

is the angle between $v$ and the reference neighbor after projection to the tangent plane.

### A.2  Parallel Transport

Let $f_p$ and $f_q$ be feature vectors at nodes $p, q$, respectively, where $q$ is adjacent to $p$. As features at different nodes live in different tangent spaces and have different bases, in order for the anisotropic kernel to be applied to $f_q$ for each $q \in \mathcal{N}_p$, each $f_q$ must be parallel transported to $T_p M$ and be in the same gauge.

Let $f_p$ and $f_q$ be feature vectors at nodes $p, q$, respectively, where $q$ is adjacent to $p$. As mentioned in the main text, the parallel transporter $g_{q \to p}$ first aligns the tangent plane $T_p M$ to the tangent plane $T_q M$ by a 3D rotation and then transforms the gauge at $q$ to the gauge at $p$ by a planar rotation.

Let $n_p, n_q$ be the normal at $p$ and $q$, and $R_{qp} \in SO(3)$ be the 3D rotation required to align $T_q M$ and $T_p M$. Then $g_{q \to p} \in [0, 2\pi)$ is given by

$$g_{q \to p} = \mathrm{atan2}\left((R_{qp} e_2^q)^\top e_1^p, (R_{qp} e_1^q)^\top e_1^p\right), \tag{9}$$

where $(e_1^q, e_2^q)$ and $(e_1^p, e_2^p)$ are the frames at node $q$ and $p$, respectively.

## B  Gauge-Equivariant Message Passing Methods

In this section, we provide the architectures of GemCNN and EMAN for the wave dataset to compare with Hermes (see Figure 2), and prove that Hermes is equivariant to local gauge transformations.

### B.1  GemCNN

The architecture for GemCNN is given in Figure 6. Note that while the equations in GemCNN (see Equations 1) allow for self-interactions, the implemented architecture does not actually use self-interaction kernels, possibly due to residual connections.

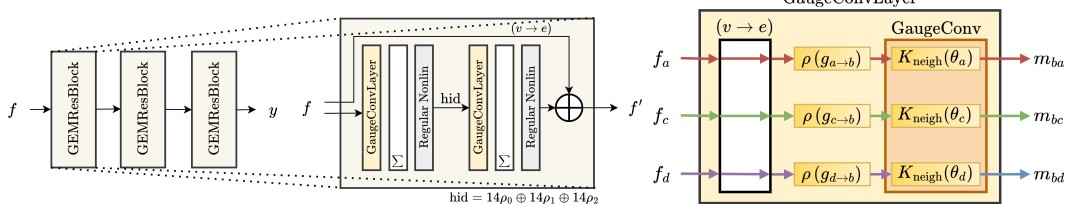

Figure 6: GemCNN network architecture for the Wave PDE dataset. There are three message blocks, each with 2 gauge-equivariant convolutional layers. To illustrate the computations within the convolutional layer, we use the example mesh from Figure 1a and show only the computations for node $b$ for clarity.

### B.2  EMAN

For completeness, the full equations for EMAN are given in Equations (10)-(12) (see also Appendix C.5 in [10]).

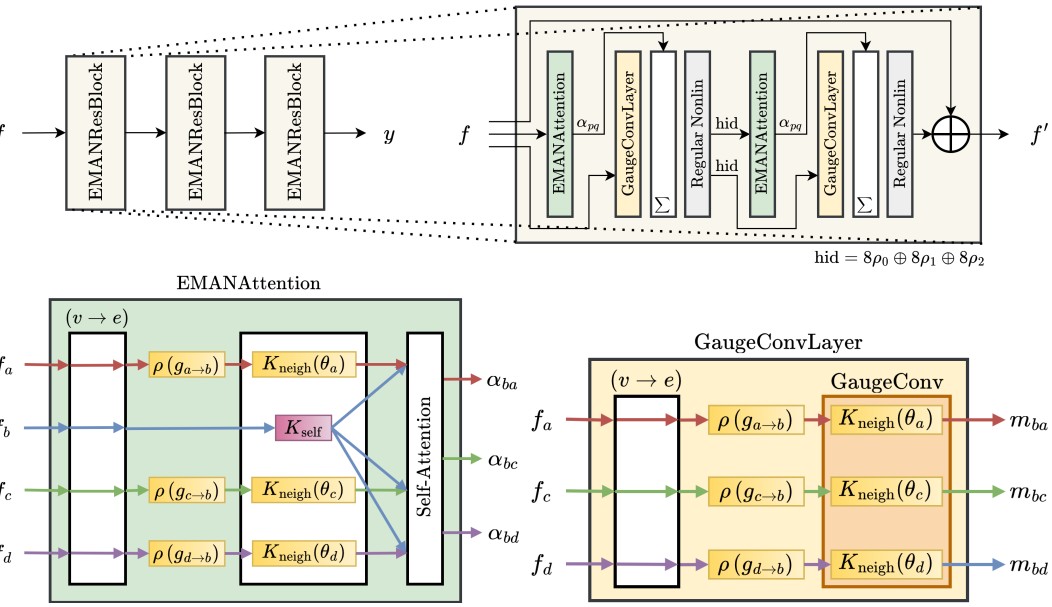

Figure 7: EMAN network architecture for the Wave PDE dataset. There are three message blocks, each with 2 attention and gauge-equivariant convolutional layers. To illustrate the computations within the edge and node networks, we use the example mesh from Figure 1a and show only the computations for node $b$ for clarity.

$$f'_p = \alpha_{pp} K^{\text{self}}_{\text{value}} f_p + \sum_{q \in \mathcal{N}_p} \alpha_{pq} K^{\text{neigh}}_{\text{value}}(\theta_q) \rho(g_{q \to p}) f_q, \tag{10}$$

$$\alpha_{pp} = \text{softmax} \left( \frac{\left( K^{\text{self}}_{\text{key}} f_p \right)^\top (K_{\text{query}} f_p)}{\sqrt{C_{\text{att}}}} \right), \tag{11}$$

$$\alpha_{pq} = \text{softmax} \left( \frac{\left( \|_{q \in \mathcal{N}_p} K^{\text{neigh}}_{\text{key}}(\theta_{pq}) \rho(g_{q \to p}) f_q \right)^\top (K_{\text{query}} f_p)}{\sqrt{C_{\text{att}}}} \right), \tag{12}$$

where $K^i_{\text{key}}, K^i_{\text{value}}, i \in \{\text{self}, \text{neigh}\}$ are the usual key and value matrices for the self-interaction and neighbor interaction terms, and $K_{\text{query}}$ is the query matrix. The $\|_{q \in \mathcal{N}_p}$ denotes a concatenation of the vectors for all $q \in \mathcal{N}_p$.

Figure 7 shows the architecture for EMAN. There is an EMANAttention layer to compute the attention weights $\alpha_{pq}$, which are passed to the aggregation function. Similarly to GemCNN, the implemented architecture does not use self-interaction kernels.

## B.3 Hermes

We provide the proof of Proposition 1 (stated here again for convenience). Part of the proof uses the fact that anisotropic convolutions over the tangent space are equivariant under the appropriate kernel constraint as shown in de Haan et al. [9].

**Proposition 2.** *Hermes is equivariant to local gauge transformations $g^p$ for any $p \in \mathcal{V}$ of a mesh.*

*Proof.* Using Equations (3)-(7), we first consider the simpler case where the edge and node networks have one gauge-equivariant layer each, i.e. $N_e = N_n = 1$. Let $G = SO(2)$ be the gauge group. Let $p \in \mathcal{V}$ be some vertex and $q \in \mathcal{N}_p$ be some vertex adjacent to $p$. The inputs to the edge

network are $f_p, f_q, e_{pq}$ (the source node, target node, and edge features, respectively), which are all representations of $G$. With respect to the gauge transformation $g^p \in G$, they transform as

$$f_p \mapsto \rho_{\text{in}}(-g^p)f_p, \quad \rho(g_{q \to p})f_q \mapsto \rho_{\text{in}}(-g^p)\rho(g_{q \to p})f_q, \quad e_{pq} \mapsto \rho_{\text{in}}(-g^p)e_{pq},$$

where $f_q$ is first parallel transported to the gauge at $p$ before applying the group action ($e_{pq}$ is assumed to already be in this gauge). As $h_{pq}$ is the direct product of $f_p, \rho(g_{q \to p})f_q, e_{pq}$ (Equation 3), it also transforms as $h_{pq} \mapsto \rho_{\text{in}}(-g^p)h_{pq}$. For the edge network to be equivariant with respect to $g^p$, we need to show that $\sigma\left(K^1_{\text{neigh}}(\theta_q - g^p)(\rho_{\text{in}}(-g^p)h_{pq})\right) = \rho_{\text{out}}(-g^p)\left(\sigma\left(K^1_{\text{neigh}}(\theta_q)h_{pq}\right)\right)$. We compute the left-hand side

$$\sigma\left(K^1_{\text{neigh}}(\theta_q - g^p)(\rho_{\text{in}}(-g^p)h_{pq})\right) = \sigma\left(\rho_{\text{out}}(-g^p)K^1_{\text{neigh}}(\theta_q)\rho_{\text{in}}(g^p)\rho_{\text{in}}(-g^p)h_{pq}\right)$$
$$= \rho_{\text{out}}(-g^p)\sigma\left(K^1_{\text{neigh}}(\theta_q)h_{pq}\right)$$

as desired, where we have used the kernel constraint $K^1_{\text{neigh}}(\theta_q - g^p) = \rho_{\text{out}}(-g^p)K^1_{\text{neigh}}(\theta_q)\rho_{\text{in}}(g_{q \to p})$ and the fact that $\sigma$ is equivariant with respect to the $\rho_{\text{out}}$ representation of $G$ and thus commutes with $\rho_{\text{out}}(-g^p)$.

For the node network, $K^1_{\text{self}}$ does not depend on $\theta_q$ and so the kernel constraint is simpler: $K^1_{\text{self}} = \rho_{\text{out}}(-g^p)K^1_{\text{self}}\rho_{\text{in}}(g^p)$. It is easy to see that $\sigma\left(K^1_{\text{self}}\rho_{\text{in}}(-g^p)h_p\right) = \rho_{\text{out}}(-g^p)\left(\sigma\left(K^1_{\text{self}}h_p\right)\right)$ and we omit the proof as it is nearly identical to that of the edge network.

The remaining part is the aggregation (sum) (Equation 5). As $\rho(-g^p)$ is a linear group action (representation) of $G$, $\rho(-g^p)m_p = \sum_{q \in \mathcal{N}_p} \rho(-g^p)m_{pq}$.

We have so far proved that Hermes is equivariant for $N_e = N_n = 1$. This suffices to prove for all $N_e \geq 1, N_n \geq 1$. The edge network $\phi_e$ is a composition of the gauge-equivariant convolutions and activations interleaved with each other. The regular nonlinearity is gauge-equivariant when the number of intermediate samples used in the discrete Fourier transform goes to $\infty$ and is approximately gauge-equivariant with finite $N$. As the composition of equivariant functions is also equivariant, $\phi_e$ is equivariant to any $g^p$ for any $N_e \geq 1$, and similarly, $\phi_n$ is equivariant to any $g^p$ for $N_n \geq 1$. Thus Hermes is equivariant to any local gauge transformation $g^p \in G$, in the limit $N \to \infty$. $\square$

## C   Experiment Domains

### C.1   Heat/Wave Equation

The heat and wave partial differential equations are given by

$$\frac{\partial u}{\partial t} = \alpha \tilde{L} u, \quad \text{(Heat)} \qquad\qquad \frac{\partial^2 u}{\partial t^2} = c^2 \tilde{L} u, \quad \text{(Wave)} \tag{13}$$

where $u$ is the temperature or the displacement for the heat and wave equations respectively, $\alpha$ is thermal diffusivity, $c$ is a constant, and $\tilde{L}$ is the symmetric cotangent Laplacian [70]. Note that $\tilde{L}$ is the discrete version of the Laplace-Beltrami operator over a Riemannian manifold. For a scalar function $u$, $\tilde{L}(u)$ is given as,

$$(\tilde{L}(u))_i = \frac{1}{2A_i} \sum_{j \in \mathcal{N}(i)} (\cot \alpha_{ij} + \cot \beta_{ij})(u_j - u_i), \tag{14}$$

where $\mathcal{N}(i)$ denotes the adjacent vertices of $i$, $\alpha_{ij}$ and $\beta_{ij}$ are the angles opposite edge $(i, j)$, and $A_i$ is the vertex area of $i$, where we use the barycentric cell area.

We use 8 example meshes from the PyVista software [30] and use 6 for training and 2 for testing generalization (test mesh) to unseen meshes. We further generate test sets to evaluate generalization on future timesteps (test time) and random initialization (test init) conditions. We simulate 5 samples for $T_{\max} = 200$ timesteps, using 3 samples from $T = 0, \ldots, 150$ for training and the remaining $T = 151, \ldots, 200$ as test time samples. The unseen 2 samples from $T = 0, \ldots, 200$ are used for test init samples. Some meshes contained too many nodes to be computationally feasible so we simplify these meshes to be less than 5,000 nodes using quadric decimation [71]. We use $\alpha = c = 1$. See Tables 4, 5 for visualizations of the meshes and how the features evolve over time.

The heat and wave equations were simulated using the finite difference method using the forward Euler scheme. As each mesh has a different Laplacian, we tune the time discretization $dt$ for each mesh. To initialize, we randomly generate Gaussian bumps over the mesh (20% of all nodes for heat, 2.5% for wave). While such initialization is not necessarily realistic, we chose to incorporate more superposition and interference to generate more diverse dynamics. The task is to predict the temperature (heat) or displacement (wave) at each vertex at the next time step, given a short history of values. Both the inputs and outputs are trivial representations.

## C.2 Cahn-Hilliard Equation

The Cahn-Hilliard equation [31] is often represented by the following two coupled second-order equations:

$$\frac{\partial c}{\partial t} - ML(\mu) = 0, \qquad\qquad \mu - \frac{df}{dc} + \lambda L(c) = 0, \qquad (15)$$

where $c$ is the fluid concentration, $M$ is the diffusion coefficient, $\mu$ is the chemical potential, $L$ is the Laplacian, $f$ is the double-well free energy function, and $\lambda$ is a positive constant.

We use 4 meshes from the PyVista software [30] and use 3 for training and 1 for testing (test mesh). As in the heat and wave datasets, we use $N = 5$ samples with $T_{\max} = 200$ timesteps and split the training and test datasets similarly. Some meshes contained too many nodes to be computationally feasible so we simplify such meshes using the same procedure as in the heat and wave datasets. See Tables 6 for visualizations.

The Cahn-Hilliard equation was simulated using the DOLFINx software [72] with time discretization $dt = 5 \times 10^{-6}$, $M = 1$, $f = 100c^2(1 - c^2)$, and we use the $\theta$-method with $\theta = 0.5$. We randomize $\lambda$ uniformly from $[0.01, 0.02]$ for each data sample. The initial concentrations are randomly sampled from $\mathcal{N}(0.6, 0.05)$ for each vertex. The solver formulates the coupled second-order PDEs in variational form, and solves it using the Newton-Krylov method [73].

The task is to predict the concentration at each vertex at the next time step, given a short history of values. Both the inputs and outputs are trivial representations.

## C.3 Shape Correspondence

The FAUST dataset [74] consists of 80 train and 20 test samples, where 2 unseen human subjects are used for testing. Following previous work [9], the inputs are $xyz$ coordinates with representation $\rho_0$ and the network outputs trivial features, followed by linear post-processing layers. The task is shape correspondence between meshes, where for every mesh, each vertex is registered and denotes the same position on the body.

## C.4 Object interactions

In this domain, we consider 50 objects on a mesh with 250 vertices, see Figure 8. Each mesh is generated parametrically using the random hills generation procedure provided in VTK [71]. For training, we generate 20 meshes using different random seeds but the same random hills parameters, randomly initialize the object positions and orientations, and then simulate for a 100 timesteps. At each timestep, we select the action with weighted probabilities: 10% turn left, 80% move forward, and 10% turn right. For testing, we generate 100 meshes with randomized hill parameters and simulate forward 20 timesteps. We represent the current state of the system using the vertex coordinates, a one-hot embedding of object occupancy, and object orientations. The orientations are 2-dimensional vectors that represent the heading of the object on the tangent place at the current vertex, pointing towards the position of the neighboring node projected onto the tangent plane.

The task is to correctly predict the object occupancy and orientations of the next timestep, given the current state and action $f(s_t, a_t) = \hat{s}_{t+1}$. The inputs to the graph network are $\rho_0$ features, except for the orientations, which are $\rho_1$. The graph network outputs both the logits of the occupancy and the orientation as $\rho_0$ and $\rho_1$ features, respectively. As the occupancy is discrete, we add a linear post-processing layer after the graph network for only the occupancy outputs and compute the negative log likelihood given the ground truth occupancy. For orientations, we use root mean square error (RMSE) from the orientations given by the graph network.

Table 4: Heat equation dataset of all meshes used with $T_{\max} = 200$. We simulate $N = 5$ samples for each mesh. Of the training meshes, we use $T = 150, \ldots, 200$ as the test time dataset and $2$ unseen samples as the test init dataset. The armadillo and urn meshes were used as the test mesh dataset.

| Object | Split | No. vertices | t=0 | t=50 | t=200 |
|---|---|---|---|---|---|
| Armadillo | Test | 1,731 | | | |
| Bunny | Train | 502 | | | |
| Lucy | Train | 2,501 | | | |
| Sphere | Train | 842 | | | |
| Spider | Train | 4,670 | | | |
| Urn | Train | 2,454 | | | |
| Woman | Train | 2,998 | | | |

Table 5: Wave equation dataset of all meshes used with $T_{max} = 200$. We simulate $N = 5$ samples for each mesh. Of the training meshes, we use $T = 150, \ldots, 200$ as the test time dataset and 2 samples as the test init dataset. The armadillo and urn meshes were used as the test mesh dataset.

| Object | Split | No. vertices | t=0 | t=50 | t=200 |
|--------|-------|--------------|-----|------|-------|
| Armadillo | Test | 1,731 | | | |
| Bunny | Train | 502 | | | |
| Lucy | Train | 2,501 | | | |
| Sphere | Train | 842 | | | |
| Spider | Train | 4,670 | | | |
| Urn | Train | 2,454 | | | |
| Woman | Train | 2,998 | | | |

Table 6: Cahn-Hilliard equation dataset of all meshes used with $T_{max} = 200$. We simulate $N = 5$ samples for each mesh. Of the training meshes, we use $T = 150, \ldots, 200$ as the test time dataset and 2 samples as the test init dataset. The armadillo and urn meshes were used as the test mesh dataset.

| Object | Split | No. vertices | T=0 | T=50 | T=200 |
|--------|-------|--------------|-----|------|-------|
| Bunny | Train | 502 | | | |
| Ellipsoid | Train | 1,942 | | | |
| Sphere | Train | 842 | | | |
| Supertoroid | Test | 2,940 | | | |

Figure 8: Objects interactions on a mesh. Purple denotes an unoccupied vertex and other colors represent different objects. Images are different random initializations of the underlying mesh and object position and orientations.

## D   Training Details and Baselines

For classification tasks such as FAUST or for predicting object occupancy in the Objects dataset, post-processing linear layers are used. For FAUST, we find that training for longer (400 epochs vs 100 from [9]) benefits all models. We use cross entropy loss for FAUST, a combination of binary cross entropy and RMSE for Objects, and RMSE for all the PDE datasets. For the PDE datasets, we use truncated backpropagation through time [75] and roll out the network predictions for 3 timesteps using a history of 5 timesteps.

We use the regular nonlinearity [9] to maintain gauge equivariance, with ReLU activations and 101 samples, for all models. We use a band limit of 4 for the gauge-equivariant kernels. Throughout,

Table 7: Details of architecture and hyperparameters

| Dataset | FAUST | Objects | Heat | Wave | Cahn-Hilliard |
|---|---|---|---|---|---|
| No. blocks | 5 | 1 | 1 | 2 | 1 |
| No. layers in $\phi_e$ per block | 2 | 3 | 4 | 2 | 4 |
| No. layers in $\phi_n$ per block | 0 | 0 | 3 | 1 | 1 |
| Hidden representation | $\oplus_{i=0}^{2} 10\rho_i$ | $\oplus_{i=0}^{2} 12\rho_i$ | $\oplus_{i=0}^{2} 12\rho_i$ | $\oplus_{i=0}^{2} 12\rho_i$ | $\oplus_{i=0}^{2} 12\rho_i$ |
| No. post processing layers | 2 | 1 | 0 | 0 | 0 |
| Epochs | 400 | 100 | 100 | 100 | 100 |
| Batch size | 1 | 2 | 1 | 1 | 1 |
| Optimizer | | | Adam | | |
| Learning rate | $7 \times 10^{-3}$ | $5 \times 10^{-4}$ | $1 \times 10^{-4}$ | $5 \times 10^{-4}$ | $5 \times 10^{-3}$ |

Table 8: Comparison table of relevant graph neural networks and their features.

| Models | Graph Flavors | | | Features | | |
|---|---|---|---|---|---|---|
| | Convolutional | Attentional | Message Passing | E(3)-Equivariant | Gauge-Equivariant | Mesh Aware |
| GemCNN | ✓ | | | | ✓ | ✓ |
| EMAN | | ✓ | | ✓ (w/ RelTan features) | ✓ | ✓ |
| Hermes | | | ✓ | | ✓ | ✓ |
| GCN | ✓ | | | | | |
| MPNN | | | ✓ | | | |
| MeshGraphNet | | | ✓ | | | ✓ |
| EGNN | | | ✓ | ✓ | | |
| SpiralNet++ | | | ✓ | | | ✓ |

Table 9: Number of learnable parameters

| | FAUST | Objects | Heat | Wave | Cahn-Hilliard | Flag |
|---|---|---|---|---|---|---|
| GemCNN | 1,829,658 | 161,715 | 40,337 | 40,337 | 29,775 | - |
| EMAN | 1,839,476 | 165,013 | 40,093 | 40,093 | 30,882 | - |
| Hermes | 1,837,776 | 162,976 | 40,544 | 40,695 | 23,122 | 51,555 |
| GCN | - | - | 41,401 | 41,401 | 29,401 | - |
| MPNN | - | - | 41,449 | 41,449 | 29,673 | - |
| MeshGraphNet | - | - | 44,993 | 44,993 | 35,457 | 45,313 |
| EGNN | - | - | 50,433 | 50,433 | 28,609 | - |
| SpiralNet++ | - | - | 63,841 | 63,841 | 27,153 | - |

L2 regularization of the weights with a coefficient of $1 \times 10^{-5}$ was used. A cosine learning rate scheduler was used for the Objects and Cahn-Hilliard datasets.

Table 7 contains the architecture details and hyperparameters used for each domain. Table 8 lists each method's flavor of message passing and its features. Table 9 shows the number of learnable parameters used for each method and domain.

# E   Results

Table 10 shows the results on FAUST, Objects, and FlagSimple datasets. On FAUST, GemCNN outperforms both EMAN and Hermes. We obtained slightly different results for EMAN than reported in the paper [10]. On Objects, Hermes outperforms GemCNN and EMAN, albeit slightly. On FlagSimple, Hermes considerably outperforms MeshGraphNet.

Table 10: Test results for each method using 5 runs. Gray numbers are $95\%$ confidence intervals.

| | | GemCNN | EMAN | Hermes | MeshGraphNet |
|---|---|---|---|---|---|
| FAUST | Accuracy (%) | **99.1**$_{\pm 0.1}$ | 97.5$_{\pm 0.4}$ | 97.5$_{\pm 0.2}$ | - |
| OBJECTS | Accuracy (%) | 98.8$_{\pm 0.1}$ | 96.9$_{\pm 0.5}$ | **99.7**$_{\pm 0.0}$ | - |
| | RMSE ($\times 10^{-2}$) | 6.40$_{\pm 0.1}$ | 6.43$_{\pm 0.0}$ | **6.26**$_{\pm 0.0}$ | - |
| FLAGSIMPLE | RMSE ($\times 10^{-3}$) | - | - | **5.87**$_{\pm 0.0}$ | 9.01$_{\pm 0.1}$ |

## E.1 Long-horizon prediction rollouts

Table 11 shows additional qualitative samples generated autoregressively using only the initial conditions for $T + 10, T + 30, T + 100$ timesteps. Hermes predicts large spatial patterns accurately and outperforms GemCNN and EMAN on all datasets.

Table 11: Qualitative predictions rolled out to $10, 30, 100$ timesteps using only initial conditions.

## E.2 Mesh Fineness

Figure 9 shows visualizations of the different mesh resolutions used and Figure 10 shows the results on the wave dataset. Hermes still outperforms GemCNN and EMAN at each resolution. Unlike the heat dataset (Figure 4), there is an unexpected overall decreasing RMSE trend with a coarser mesh.

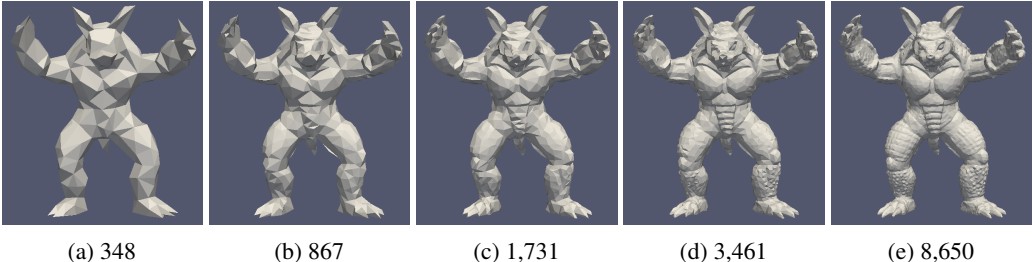

(a) 348     (b) 867     (c) 1,731     (d) 3,461     (e) 8,650

Figure 9: Visualization of different mesh resolutions. The captions denote the number of vertices.

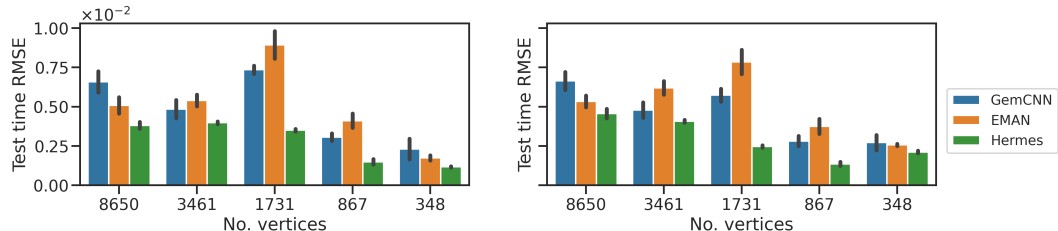

Figure 10: Performance for varying mesh fineness for wave on the test time (left) and test init (right) datasets. Error bars denote standard error over 3 runs. The number of vertices decreases with an increasing reduction ratio (see Figure 9).

## E.3 Surface Roughness

Figure 11 shows visualizations of the different mesh roughness and Figure 12 shows the results on the wave dataset. Hermes outperforms GemCNN and EMAN at most roughness scales.

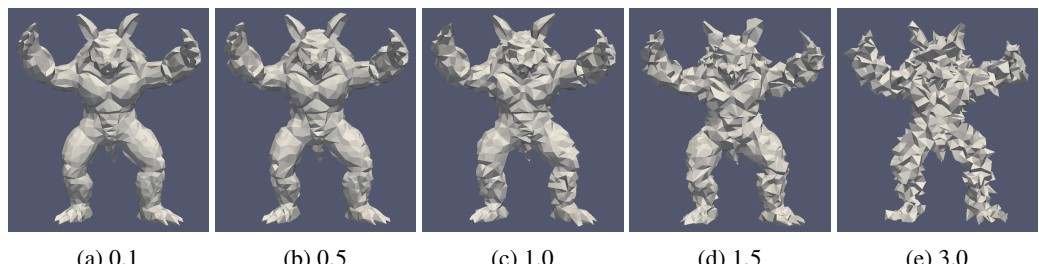

(a) 0.1     (b) 0.5     (c) 1.0     (d) 1.5     (e) 3.0

Figure 11: Visualization of different mesh roughnesses. The captions denote the roughness scale.

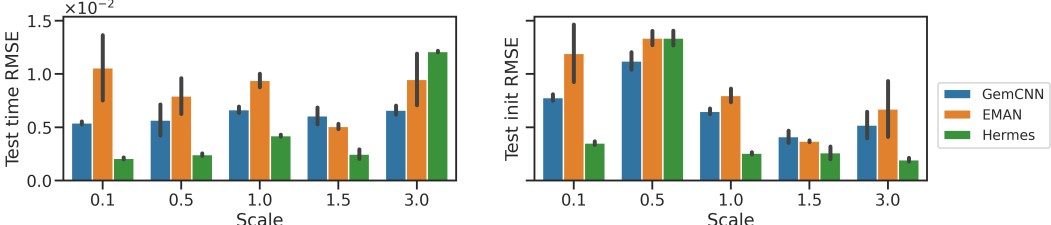

Figure 12: Performance for varying surface roughness for wave on the test time (left) and test init (right) datasets. Error bars denote standard error over 3 runs. Increasing scale increases the surface roughness of the mesh (see Figure 11).

### E.4 Ablation on Residual Connection

Table 12 shows the ablation experiment with the residual connection. The results are mixed: having a residual helps on Heat but not on Wave or Cahn-Hilliard. The decision to have a residual connection should thus be considered a task-dependent hyperparameter.

Table 12: Ablation study on residual connection

|  |  | With residual | Without residual |
|---|---|---|---|
| HEAT | Test time ($\times 10^{-3}$) | $1.18_{\pm 0.3}$ | $1.24_{\pm 0.5}$ |
|  | Test init ($\times 10^{-3}$) | $1.16_{\pm 0.3}$ | $1.22_{\pm 0.5}$ |
|  | Test mesh ($\times 10^{-3}$) | $1.01_{\pm 0.3}$ | $1.05_{\pm 0.4}$ |
| WAVE | Test time ($\times 10^{-3}$) | $5.43_{\pm 0.8}$ | $5.17_{\pm 0.6}$ |
|  | Test init ($\times 10^{-3}$) | $3.72_{\pm 1.3}$ | $2.38_{\pm 0.3}$ |
|  | Test mesh ($\times 10^{-3}$) | $3.79_{\pm 1.3}$ | $2.52_{\pm 0.4}$ |
| CAHN-HILLIARD | Test time ($\times 10^{-3}$) | $4.23_{\pm 0.9}$ | $3.08_{\pm 0.8}$ |
|  | Test init ($\times 10^{-3}$) | $5.21_{\pm 1.2}$ | $3.53_{\pm 1.3}$ |
|  | Test mesh ($\times 10^{-3}$) | $5.34_{\pm 0.7}$ | $5.73_{\pm 1.2}$ |

