# OpenReview forum: "Modeling Dynamics over Meshes with Gauge Equivariant Nonlinear Message Passing"
_NeurIPS.cc/2023/Conference — NeurIPS 2023 poster_

### Official Review · Reviewer_P6sJ · 2023-06-16

**Soundness:** 2 fair
**Presentation:** 3 good
**Contribution:** 2 fair
**Rating:** 3
**Confidence:** 5

**Summary:**

This paper studies the problem of gauge equivariant convolutional and attentional architectures on meshes and proposes to introduce non-linear activations to enhance the model. The experiments on three models shows the performance of the proposed method.

**Strengths:**

S1. The studied problem is important.

S2. The presentation is good.

S3. Equivariance is an important property in message passing neural networks.

**Weaknesses:**

W1. The most important point is the novelty. Actually, Equivariant Mesh Attention Networks have combined Gauge Equivariance with MPNN. Your work adds non-linear activation [16] into the original models. The technical contribution is week.

W2. Given that your claim that "the combination of nonlinear message passing and gauge equivariance has not been proposed", you should introduce "nonlinear" into the title and give more emphasis on the significance of it.

W3. Besides the performance, I didn't see too many deep insights about the benefit of nonlinear terms. For example, how to choose the non-linear activation? How the "non-linear" activation function influence the solution of non-linear PDE function. Since the difference of our method is minor, you should introduce deep insights to strength your contribution.

W4. How about add non-linear activation in different places? I hope to get some deep results.

W5. The performance of your methods seems to not good on FAUST used in EMAN. Why you skip TOSCA? Can your method compared with [R1,R2,R3] on some standard benchmarks reported in [R1,R2,R3].


[R1] Learning Mesh-Based Simulation with Graph Networks, ICLR 21

[R2] EAGLE- Large-scale Learning of Turbulent Fluid Dynamics with Mesh Transformers, ICLR 23

[R3] Predicting Physics in Mesh-reduced Space with Temporal Attention, ICLR 22.


**Questions:**

See weaknesses.

**Limitations:**

See weaknesses.

---

> ### Author Rebuttal · Authors · 2023-08-10
>
> We thank you for the detailed feedback and hope to have addressed all of your concerns.
>
> > The most important point is the novelty.
>
> We respectfully disagree that our work only differs from [16] by adding a non-linear activation. Our work explores non-linear message passing with gauge-equivariance on meshes, an important design decision to consider with respect to the three flavors of message passing. Previous works have highlighted the importance of nonlinear message passing for modeling complex interactions [A, B] and also for equivariant graph networks [C]. It is thus natural to explore how nonlinear message passing would benefit gauge-equivariant methods. There is also a clear conceptual difference to Hermes than simply adding a nonlinearity to GemCNN. Nonlinear message passing effectively decouples the hop distance in the graph from the number of nonlinear layers and gives practitioners the ability to choose the receptive field freely from the network depth.  This makes the input graph and computational different which may be useful for certain tasks. Throughout the literature, various methods have been proposed to perform this decoupling such as graph sparsification [D], diffusion [E], or dynamic rewiring [F]. In our work, we demonstrate that this decoupling may be crucial in modeling complex interactions and surface dynamics on meshes.
>
> > Given that your claim that "the combination of nonlinear message passing and gauge equivariance has not been proposed", you should introduce "nonlinear" into the title
>
> We agree and we will include the word “nonlinear” in the title of the final version.  We adopted our terminology from [G] which labels the three flavors of GNN as convolutional, attentional, or message-passing.  Sometimes the last is also referred to as “general message passing,” and we tried to be consistent with the literature.
>
> > Besides the performance, I didn't see too many deep insights about the benefit of nonlinear terms.
>
> We emphasize that the contribution of our method is not the addition of specific nonlinearity, but rather the proposal of nonlinear functions for the message and update functions to compute nonlinear interaction messages in gauge-equivariant networks. This stems from an important design decision to compute nonlinear interactions between vertices, which we demonstrate to be important in predicting complex surface dynamics. Furthermore, this also decouples one axis of architecture design (network depth) from the receptive field of the network, which is important in improving model performance without additional learnable parameters. The specific nonlinearity used is a hyperparameter in our method, dependent on the task.
>
> >How about add non-linear activation in different places?
>
> We point out that adding non-linear activation in different places does not change the nonlinearity of the message and update functions. The “degree” of nonlinearity depends on the network depth and so nonlinear activations should be interspersed with the convolution layers in the architecture. Thus the axis to consider is the number of layers in the message and update functions. In all datasets, we tuned the Hermes architecture so that the message and update functions consist of a different number of layers and activations.
>
> >Why you skip TOSCA? Can your method compared with [R1,R2,R3] on some standard benchmarks reported in [R1,R2,R3].
>
> We were unable to obtain the TOSCA dataset as the public download URL is down. We clarify that we are not chasing state-of-the-art results but propose a new architecture that combines nonlinear message passing with gauge equivariance for meshes. We demonstrate the importance of computing nonlinear messages in predicting complex surface dynamics. However, we agree that having more baselines would be beneficial. We have included MeshGraphNet [A] as a baseline and also add two non-equivariant, non mesh-aware baselines (GCN and MPNN), an E(3)-equivariant non mesh-aware baseline (EGNN), and a non-equivariant, mesh-aware method (SpiralNet++). All methods use a similar number of parameters. See Table 1 in the uploaded figures/tables page for the comparison of features of each method. Results show that Hermes outperforms other baselines on all tasks, with the exception of MeshGraphNet on the Heat dataset. Interestingly, Hermes significantly outperforms MeshGraphNet on the test mesh dataset for Wave and Cahn-Hilliard, suggesting that Hermes can more accurately learn the dynamics function without being mesh-specific.
>
> We also add the FlagSimple dataset from [A] and the results show that Hermes outperforms MeshGraphNet. We will include these additional results in the final version.
>
> References:
> - [A] Battaglia, P., Pascanu, R., Lai, M., & Jimenez Rezende, D. (2016). Interaction networks for learning about objects, relations and physics. Advances in neural information processing systems.
> - [B] Kipf, T., Fetaya, E., Wang, K. C., Welling, M., & Zemel, R. (2018). Neural relational inference for interacting systems. In International conference on machine learning.
> - [C] Brandstetter, J., Hesselink, R., van der Pol, E., Bekkers, E. J., & Welling, M. (2021). Geometric and Physical Quantities improve E (3) Equivariant Message Passing. In International Conference on Learning Representations.
> - [D] Hamilton, W., Ying, Z., & Leskovec, J. (2017). Inductive representation learning on large graphs. Advances in neural information processing systems.
> - [E] Gasteiger, J., Weißenberger, S., & Günnemann, S. (2019). Diffusion improves graph learning. Advances in neural information processing systems.
> - [F] Gutteridge, B., Dong, X., Bronstein, M. M., & Di Giovanni, F. (2023). DRew: Dynamically Rewired Message Passing with Delay. In International Conference on Machine Learning.
> - [G] Bronstein, M. M., Bruna, J., Cohen, T., & Veličković, P. (2021). Geometric deep learning: Grids, groups, graphs, geodesics, and gauges.

---

> > ### Comment · Reviewer_P6sJ · 2023-08-18
> > **Thanks for your response.**
> >
> > Thanks for your response. First, can you please summarize your technical contribution about the model and make a detailed comparison with the related works? To me, decoupling the hop distance is still kind of trivial in graph communities. Second, nonlinear functions for the message and update functions can date from [1], and a range of graph methods work on this, which is usually considered implementation details in some recent works. Third, since NeurIPS is the best venue in machine learning, I suggest to do more comparison with more state-of-the-arts.
> >
> > [1] Simplifying Graph Convolutional Networks, ICML 19.

---

> > > ### Author Response · Authors · 2023-08-20
> > >
> > > Thank you for your reply. Please see our response below.
> > >
> > > > can you please summarize your technical contribution about the model and make a detailed comparison with the related works?
> > >
> > > Our main contribution is proposing a novel method that combines gauge-equivariance and nonlinear message-passing. While both convolutional and attentional gauge-equivariant architectures have been introduced (GemCNN and EMAN), such methods can only linearly approximate the interactions between node neighbors, i.e. linear message passing. Hermes generalizes these to pass nonlinear messages. Modifying GemCNN and EMAN to pass nonlinear messages requires a more conceptual change than simply adding a nonlinearity after the layer: one needs to use nonlinear function approximators such as neural networks for the node and edge networks within the layer. Nonlinear message passing has been shown to be essential in object-relational reasoning and physics environments [A, B, C] and this is a central contribution of these papers. We specifically show how nonlinear gauge-equivariant message passing is more beneficial in modeling complex surface dynamics over convolutional and attentional versions. Additionally, we note that making message passing networks gauge-equivariant is non-trivial and a novel contribution. The objective of our paper is not to create a new architecture to be the state-of-the-art in standard benchmarks, but rather to extend the design space of neural networks for meshes (particularly with respect to equivariant networks) and to add insights about in which scenarios nonlinear message passing should be used.
> > >
> > > References:
> > > - [A] Battaglia, P., Pascanu, R., Lai, M., & Jimenez Rezende, D. (2016). Interaction networks for learning about objects, relations and physics. Advances in neural information processing systems.
> > > - [B] Kipf, T., Fetaya, E., Wang, K. C., Welling, M., & Zemel, R. (2018). Neural relational inference for interacting systems. In International conference on machine learning.
> > > - [C] Brandstetter, J., Hesselink, R., van der Pol, E., Bekkers, E. J., & Welling, M. (2021). Geometric and Physical Quantities improve E (3) Equivariant Message Passing. In International Conference on Learning Representations.
> > >
> > > > nonlinear functions for the message and update functions can date from [1], and a range of graph methods work on this, which is usually considered implementation details in some recent works
> > >
> > > If we understand correctly, the reference [1] removes nonlinearities from GCN layers and combine the weights into a single linear layer called SGC. A key difference between the topic of [1] and our paper is that [1] removes nonlinearities from GCN layers where the aggregation step is performed directly after the convolution and before the nonlinearity. Thus both GCN layers and their proposed SGC use only linear messages. In our paper, we specifically use nonlinear functions before the aggregation step, leading to nonlinear message passing.
> > >
> > > > I suggest to do more comparison with more state-of-the-arts.
> > >
> > > We have added several new baselines including a state-of-the-art (SOTA) mesh method (MeshGraphNet), a popular and standard mesh method (SpiralNet++), and also a SOTA E(3)-equivariant method (EGNN). We aimed to compare against baselines along several different design axes, to assess how helpful gauge equivariance or nonlinear message passing is with respect to modeling dynamics. We note that our goal was not to introduce a new SOTA method but rather to investigate the three flavors of message passing and when nonlinear message passing works better.

---

### Official Review · Reviewer_CzKN · 2023-07-04

**Soundness:** 3 good
**Presentation:** 4 excellent
**Contribution:** 3 good
**Rating:** 6
**Confidence:** 2

**Summary:**

The authors describe a message-passing mesh-based gauge equivariant architecture. The architecture is described as the natural follow-up from previous equivariant architecture. In short, an edge network aggregates information between source, target nodes, and edge features. Then, these "messages" are aggregated with a gauge equivariant convolution to the target node, accounting also for self-interactions.
Such architecture is tested on 4 applications: shape correspondences, object interactions, PDE on meshes, modeling heat eq., and Cahn-Hilliard equation on a mesh. The authors compare it with other message-passing architecture proving the soundness and effectiveness of the technique.

**Strengths:**

The described architecture is the natural follow-up from previous works (as the authors describe it). Its description and illustration are well crafted, and the text is clear and well-structured. The experiments are properly designed to show the benefits of this approach with respect to previous methods, hence showing numerical and qualitative improvements. Furthermore, the authors address important concerns such as mesh finesses and roughness.

**Weaknesses:**

The main weakness of the method is the requirement of a reasonable topology, which cannot be overcome. This may limit severely the application of these types of methods.
Other than the above, as stated by the authors the method is relatively slow (or slower) compared to baselines (eg GemCNN).

**Questions:**

Overall I am quite happy with the paper, there are a few concerns I wish the authors would address:
- has this method been tested on incredibly large meshes (>1M vertices)?
- I would be interested to know what are the performance be wrt a method like MeshCNN in applications like mesh classification? both in terms of time and accuracy. (if the authors know this it would be great)
- based on the visual results, it looks like the long-term roll out is far from GT, do the authors have any idea on how to fix this?
- would alternating different message passing layers (conv->hermes->...) help in terms of speed up without excessively harming the performances?

**Limitations:**

I do not see any negative societal impact.

I would recommend the authors add in the next submission (if any) other tasks to the paper such as mesh segmentation and classification and compare with know baselines. This should show the strength of the method. It would also be interesting to assess the robustness of the method with corrupted topologies, this could be also an interesting point for possible follow-up.

---

> ### Author Rebuttal · Authors · 2023-08-10
>
> Thank you for the detailed comments and positive feedback.
> > The main weakness of the method is the requirement of a reasonable topology ... the method is relatively slow (or slower) compared to baselines (eg GemCNN).
>
> We respectfully argue that meshes can approximate any Riemannian manifold, which covers many possible surfaces and objects encountered in the real world. Furthermore, most objects can be approximated as manifold meshes where there are no singularities or unreasonable boundary conditions. Many previous mesh-based methods assume manifold meshes [A, B, C], including all 3 gauge-equivariant methods (GemCNN, EMAN, Hermes). We feel that the assumption of a manifold mesh is not a severe limitation.To handle unreasonable topologies or non-manifold meshes, one could perform a classing meshing technique [D] to make it a manifold. It may also be possible to modify Hermes to handle such meshes by not using gauge equivariance on the problematic vertices and edges.
>
> Regarding the computation time, we recorded the forward computation time of each model during inference on the test time dataset (Table 3 of the tables page). We find that Hermes actually has lower computation time in the forward pass than GemCNN, due to the fact that it performs a smaller number of aggregations than GemCNN and with a similar number of parameters. We will adjust the discussion to reflect this.
> > has this method been tested on incredibly large meshes (>1M vertices)?
>
> We restricted our experiments to meshes with up to 4670 vertices. As meshes inherently contain more data than other 3D representations such as point clouds or voxels, they require more memory to process. Though we simplify this and process meshes as graphs, even graph neural networks cannot scale to large graphs due to memory constraints [E, F]. We are not aware of any message passing graph networks that can scale up to 1M vertices on a single GPU and scaling to such large meshes would likely require significant distributed training and/or multi-scale methods with graph subsampling. It would be a good avenue to explore to apply such techniques to Hermes.
> > ... the performance be wrt a method like MeshCNN in applications like mesh classification? both in terms of time and accuracy.
>
> As a nonlinear message passing and gauge-equivariant method, Hermes is more suited to predicting complicated interactions and irregular, rough meshes, which is supported by the performance increase on surface PDE datasets. In mesh classification, it seems that it is more important to recognize local shapes and surface curvatures and we hypothesize that interactions between neighboring vertices would likely be close to linear. Methods such as MeshCNN would likely perform better in these scenarios. Regarding computation time, all three flavors of gauge-equivariant methods require more computation time than regular graph neural networks. As MeshCNN effectively performs a convolution over the mesh faces, it would likely be more computationally efficient than either GemCNN or Hermes, which perform convolutions over edges.
> > ... it looks like the long-term roll out is far from GT
>
> We would argue that the long-term rollouts for Hermes do not look that far from GT as the local patches seem to match GT in size and sign, though the magnitudes may differ slightly. One possible way to improve performance is to use normalization for the node and edge input features or to predict differentials (e.g. x_t+1 - x_t) as targets as done in [3]. Another way could be to use multi-step predictions in the training loss. We note that these techniques are general and not specific to our method, and performance may depend on the task.
> >would alternating different message passing layers (conv->hermes->...) help in terms of speed up without excessively harming the performances?
>
> It’s an interesting idea! It would combine the parameter/computational efficiency of the convolution but also retain the ability to compute nonlinear messages. One could also use the gauge-equivariant convolution from GemCNN to preserve gauge-equivariance for the entire model.
> >... other tasks to the paper such as mesh segmentation and classification and compare with know baselines ... assess the robustness of the method with corrupted topologies.
>
> As mentioned above, mesh segmentation and classification tasks are likely not suited for our method. Node correspondence results on FAUST seem to support this. We do agree that adding more baselines and datasets would be useful and so we include several more baselines (GCN, MPNN, MeshGraphNet, EGNN, SpiralNet++)
>
> We agree that experimenting with different corrupted topologies would strengthen the paper. One could also consider non-manifold meshes, with open geometries, internal faces, or non-manifold edges or vertices where we previously mentioned some possible techniques to process such meshes.
>
> References
> - [A] Hanocka, R., et al. (2019). Meshcnn: a network with an edge. ACM Transactions on Graphics.
> - [B] Pfaff, T., et al. (2020). Learning Mesh-Based Simulation with Graph Networks. In International Conference on Learning Representations.
> - [C] De Haan, P., et al. 2020. “Gauge Equivariant Mesh CNNs: Anisotropic Convolutions on Geometric Graphs.”
> - [D] Shimada, K., & Gossard, D. C. (1995). Bubble mesh: automated triangular meshing of non-manifold geometry by sphere packing. In Proceedings of the third ACM symposium on Solid modeling and applications.
> - [E] Wu, Z., et al. (2020). A comprehensive survey on graph neural networks. IEEE transactions on neural networks and learning systems.
> - [F] Duan, K., et al. (2022). A comprehensive study on large-scale graph training: Benchmarking and rethinking. Advances in Neural Information Processing Systems.

---

> > ### Comment · Reviewer_CzKN · 2023-08-13
> >
> > Thank you for taking the time to in-depth respond to my concern and for the additional evaluation.
> >
> > ### Topology
> > I agree meshes can approximate any surface my point was more related to the manifoldness and self-intersection of the mesh itself. Your response addressed my concerns.
> >
> > ### 1M vertices
> > I understand handling such large meshes is difficult, I was wondering if that is the case for this method. Thank you for your response.
> >
> > ### Mesh classification
> > Interesting, I understand the inductive bias required to perform mesh classification it is different than PDEs, and I wonder if authors have any insight/intuition on how to change the current pipeline to adapt it to such tasks? Perhaps with informed vertex clustering to massively reduce the mesh resolution?
> >
> > ### Roll out
> > I disagree, based on visual analysis the results are on average close but quite different in details. Probably the residual approach would further improve the result. Thank you for the response.

---

> > > ### Author Response · Authors · 2023-08-14
> > >
> > > For a global-level task such as mesh classification (compared to node classification), modeling long-range interactions between local shape structures on the object is likely key. Informed vertex clustering such that each cluster represents local shapes would be a great approach. One could perform the clustering in a non-learned manner using additional information about the object structure, such as its topology or other global features. One could also use a learned approach by using multi-scale or hierarchical methods. At coarse scales, different local basic shapes on the object that are far away from each other could interact and pass messages, while the finer features within the local shapes could be handled at higher resolutions. We note that this idea has been proven successful for the analogous task of graph classification [A, B, C].
> > >
> > > References:
> > >
> > > - [A] Ying, Z., You, J., Morris, C., Ren, X., Hamilton, W., & Leskovec, J. (2018). Hierarchical graph representation learning with differentiable pooling. Advances in neural information processing systems, 31.
> > > - [B] Lee, J., Lee, I., & Kang, J. (2019, May). Self-attention graph pooling. In International conference on machine learning (pp. 3734-3743). PMLR.
> > > - [C] Wu, Z., Jain, P., Wright, M., Mirhoseini, A., Gonzalez, J. E., & Stoica, I. (2021). Representing long-range context for graph neural networks with global attention. Advances in Neural Information Processing Systems, 34, 13266-13279.

---

### Official Review · Reviewer_yPBJ · 2023-07-05

**Soundness:** 3 good
**Presentation:** 3 good
**Contribution:** 2 fair
**Rating:** 7
**Confidence:** 4

**Summary:**

The authors propose a gauge equivariant method for simulating PDEs on the surface of meshes. Different from the convolutional and attentional prior works, the authors use non-linear message passing with gauge equivariant layers. They compare to the prior works in the FAUST shape classification and the simulation of three PDEs. They find that the convolutional method is best for the shape classification, while on the PDEs, their method achieves the best results.

**Strengths:**

- It's great to see the code included
- I agree with the authors that it's good to have non-linear message passing as an additional gauge-equivariant method.
- The paper is clearly written.

**Weaknesses:**

- The authors should include a reference to [1], which simulates fluid dynamics with gauge equivariant methods.
- It appears like the PDE experiments merely using the scalar features without information on geometry. This seems insufficient. Both [1] and [2] have suggestions for gauge equivariant geometric input features.
- The paper would benefit from an evaluation on more problems, as well as a comparison to pointcloud based methods (as done in [1]).



----------
The weaknesses have been sufficiently addressed by the rebuttal. I increase my score.

**Questions:**

- In [3], it is noted in a footnote that the kernel could be made dependent on the radius, but that the authors found it not beneficial. The method proposed here seems to also have a kernel insensitive to distance. Have the authors verified whether this is still the best choice?
- Could the authors clarify in their manuscript that the edge feature $e_{pq}$ is a feature situated on the fiber at $p$ ?
- Could the authors add a discussion of the computational cost of the various method in their paper?
- The regular non-linearity used in [3] is only approximately equivariant. Could the authors clarify that in their paper?

References:
- [1] Suk, Julian, Pim de Haan, Phillip Lippe, Christoph Brune, and Jelmer M. Wolterink. 2022. “Mesh Neural Networks for SE(3)-Equivariant Hemodynamics Estimation on the Artery Wall.”
- [2] Basu, Sourya, Jose Gallego-Posada, Francesco Viganò, James Rowbottom, and Taco Cohen. 2022. “Equivariant Mesh Attention Networks.”
- [3] De Haan, P., M. Weiler, T. Cohen, and M. Welling. 2020. “Gauge Equivariant Mesh CNNs: Anisotropic Convolutions on Geometric Graphs.”

**Limitations:**

The authors fairly reflect the limitations of their work.

---

> ### Author Rebuttal · Authors · 2023-08-10
>
> We thank the reviewer for the positive and thoughtful review. Please see our response below.
> >The authors should include a reference to [1], which simulates fluid dynamics with gauge equivariant methods.
>
> Thank you for pointing out this paper. It is indeed relevant and we will include it in the final version.
> >It appears like the PDE experiments merely using the scalar features without information on geometry. This seems insufficient. Both [1] and [2] have suggestions for gauge equivariant geometric input features.
>
> We tried using the relative tangent features proposed in [2] and observe slightly worse performance. We note that although the input/output features are scalars in the PDE datasets, the intermediate latent features are not scalars and consider irreducible representations up to order 2. In both [1] and [2], equivariant input features are constructed using either the vertex normals and/or the relative distance vector or their projections onto the tangent plane at the source node. These features are already incorporated into our model: vertex normals are used to achieve gauge equivariance and positions are used as inputs. Thus the proposed equivariant geometric input features do not necessarily convey additional information and an expressive model should be able to learn such equivariant features from data. This is rather a question of feature engineering and we agree that incorporating these features may make it easier for the model to learn the task. However, we choose the simpler option and let the model learn such features on its own, without withholding any additional information.
> >The paper would benefit from an evaluation on more problems, as well as a comparison to pointcloud based methods (as done in [1]).
>
> We agree that more evaluation would be beneficial and add the FlagSimple dataset from [A] where the task is to predict the positions of a flag blowing in the wind. We do not perform normalization or noisy training for simplicity. On this dataset, Hermes outperforms MeshGraphNet on the test dataset for this task (see Table 4 of uploaded page).
>
> Additionally, we experiment with more baselines. Specifically we consider two non-equivariant, non mesh-aware baselines (GCN and MPNN), a SOTA non-equivariant, mesh-aware method (MeshGraphNet [A]), an E(3)-equivariant non mesh-aware baseline (EGNN [B]), and a non-equivariant, mesh-aware method (SpiralNet++ [C]). EGNN can be considered a point cloud based method as it can infer edges. See Table 1 in the uploaded tables page for the comparison of features of each method and Table 2 for the results. Hermes outperforms baselines in most settings and does slightly worse than MeshGraphNets on Heat.  We will include these results in the final version.
> >In [3], it is noted in a footnote that the kernel could be made dependent on the radius, but that the authors found it not beneficial. The method proposed here seems to also have a kernel insensitive to distance. Have the authors verified whether this is still the best choice?
>
> Yes, the kernel used here is also independent of the radius. As the datasets contain generally homogeneous graphs with similar edge distances, we surmise that having a radius-dependent kernel would not change much and do not include it for computational efficiency. As our method easily accommodates edge features, we can make the messages radius dependent by adding the relative position vector and its distance as an edge feature. We tried this, but the performance was similar. For the FlagSimple dataset, we do use the edge features as described in [A].
> >Could the authors clarify in their manuscript that the edge feature e_pq is a feature situated on the fiber at p?
>
> Yes, we will clarify this point in the manuscript. We do this because edge features are often vectors relative to p and/or the edge distance which is not necessarily tied to the geometry of the edge. One could conceivably use the fiber at q or use the midpoint between p and q and parallel transport the edge features accordingly, but we choose the simpler option. We note that we did not see much performance improvement by adding edge features to the PDE datasets.
> >Could the authors add a discussion of the computational cost of the various method in their paper?
>
> We include the forward computation time during inference on the test time dataset in Table 4 of the additional page. Surprisingly, we find that Hermes is slightly faster computation-wise than GemCNN as we generally use a small number of message passing layers and use a similar number parameters. As expected, all 3 gauge-equivariant methods are significantly more computationally expensive than standard graph neural networks.
> >The regular non-linearity used in [3] is only approximately equivariant. Could the authors clarify that in their paper?
>
> We will clarify in Section 3 and in Proposition 1 that the regular nonlinearity is only equivariant in the limit when the number of samples goes to infinity. In experiments, we used an increased number of intermediate samples (101 vs. 7) in each regular nonlinearity compared to [3], leading to an equivariance error to random gauge transformations of approximately 1e-5 for the entire model.
>
> References:
> - [A] Pfaff, T., Fortunato, M., Sanchez-Gonzalez, A., & Battaglia, P. (2020, October). Learning Mesh-Based Simulation with Graph Networks. In International Conference on Learning Representations.
> - [B] Satorras, V. G., Hoogeboom, E., & Welling, M. (2021, July). E (n) equivariant graph neural networks. In International conference on machine learning (pp. 9323-9332). PMLR.
> - [C] Gong, S., Chen, L., Bronstein, M., & Zafeiriou, S. (2019). Spiralnet++: A fast and highly efficient mesh convolution operator. In Proceedings of the IEEE/CVF international conference on computer vision workshops (pp. 0-0).

---

> > ### Comment · Reviewer_yPBJ · 2023-08-11
> > **Better experiments, increased score**
> >
> > I thank the authors for their rebuttal. With their additional experiments, I will raise my score.
> >
> > There's one thing it'd like to clarify though.
> >
> > > Thus the proposed equivariant geometric input features do not necessarily convey additional information and an expressive model should be able to learn such equivariant features from data.
> >
> > I disagree. For example, a cylinder is a flat manifold. Only the global topology indicates it is different from the plane and global topology might be hard for a convolution to detect. In general, the local surface geometry (intrinsic curvature) - which is all a that a gauge equivariant method sees - might not inform how this manifold is embedded in $\mathbb R^3$ (extrinsic curvature). Similarly, in the 1D case, any curve has a flat 1D geometry, thus a 1D gauge equivariant method can not detect whether the line is embedded in the ambient space flat or curved. Additional features can inform the gauge equivariant method about the embedding and address this.

---

> > > ### Author Response · Authors · 2023-08-12
> > >
> > > We thank the reviewer for raising their score and we are pleased to have answered most questions.
> > >
> > > It’s a good point that the proposed equivariant input features in both [A, B], which use either signals based on the relative position/distance between neighbors or vertex normals, may convey extrinsic information related to the global embedding and thus may be useful for certain tasks. While these features are local, they convey information regarding the embedding that may make it easier to understand the global topology.
> > >
> > > In order to provide more direct global topological information, another easy way would be to include the absolute node positions as inputs, as done in the FAUST experiments in GemCNN and also our experiments on FAUST and Objects. (This does sacrifice E(3)-equivariance, however.) Another way would be to ``rewire'' the mesh [C] or create edges (and faces) to nodes that are close in the embedded space [D].
> > >
> > > In our experiments, the PDEs are inherently based on intrinsic local geometry and so extrinsic global information may not be as important. Indeed when we added absolute positions as inputs for the PDE datasets, we did not see much improvement in the results, likely because local dynamics dominate in these PDEs.
> > >
> > > References:
> > >
> > > - [A] Suk, Julian, Pim de Haan, Phillip Lippe, Christoph Brune, and Jelmer M. Wolterink. 2022. “Mesh Neural Networks for SE(3)-Equivariant Hemodynamics Estimation on the Artery Wall.”
> > > - [B] Basu, Sourya, Jose Gallego-Posada, Francesco Viganò, James Rowbottom, and Taco Cohen. 2022. “Equivariant Mesh Attention Networks.”
> > > - [C] Gutteridge, B., Dong, X., Bronstein, M. M., & Di Giovanni, F. (2023). DRew: Dynamically Rewired Message Passing with Delay. In International Conference on Machine Learning.
> > > - [D] Pfaff, T., et al. (2020). Learning Mesh-Based Simulation with Graph Networks. In International Conference on Learning Representations.

---

> > > > ### Comment · Reviewer_yPBJ · 2023-08-12
> > > >
> > > > That's a good point, I agree that indeed here  only the intrinsic geometry is of relevance, so possibly no additional geometric features are necessary - though I wouldn't be surprised if there are intrinsic PDE modelling cases in which they still are beneficial.
> > > >
> > > > Thanks for the discussion!

---

> > > > > ### Author Response · Authors · 2023-08-13
> > > > >
> > > > > Thank you for the helpful feedback and discussion!

---

### Official Review · Reviewer_ASko · 2023-07-06

**Soundness:** 2 fair
**Presentation:** 3 good
**Contribution:** 2 fair
**Rating:** 5
**Confidence:** 3

**Summary:**

This paper aims at solving complex partial differential equations on surfaces. Given the fact that most existing work neither incorporate surface geometry nor consider local gauge symmetries of the manifolds, this paper proposes a novel gauge equivariant network, known as Hermes, that can achieve higher performance than existing convolutional or attentional networks in certain cases. In addition, authors investigate in which cases their method has advantages over other methods.

**Strengths:**

- Propose a new gauge equivariant network, Hermes, for learning signal on meshes.
- Hermes outperforms both convolutional and attentional architectures on complex and nonlinear dynamics such as surface PDEs.
- Authors investigate in which situations nonlinear message passing should be preferred over convolutional or attentional counterparts.


**Weaknesses:**

- It is not clear what is the relationship between Hermes and previous methods that use graph networks to perform mesh-based simulation, such as [1]. Could authors elaborate on the differences with the method in [1]? And I would like to see the performance comparison between Hermes and [1] in some scenarios, for example, FlagDynamic and CylinderFlow in [1].
- In addition, Geo-FNO [2] can also perform PDE learning on irregular geometries, what is the advantages of Hermes over Geo-FNO in terms of accuracy in solving PDE on irregular domains.


[1] Pfaff, Tobias, et al. "Learning mesh-based simulation with graph networks." arXiv preprint arXiv:2010.03409 (2020).

[2] Li, Zongyi, et al. "Fourier neural operator with learned deformations for pdes on general geometries." arXiv preprint arXiv:2207.05209 (2022).

**Questions:**

- See the weakness part. I would like to raise the score if the authors can well address my concerns.

**Limitations:**

The limitations of this paper are well discussed.

---

> ### Author Rebuttal · Authors · 2023-08-10
>
> We thank the reviewer for the helpful references and hope to have addressed all of your questions.
>
> >It is not clear what is the relationship between Hermes and previous methods that use graph networks to perform mesh-based simulation, such as [1]. Could authors elaborate on the differences with the method in [1]? And I would like to see the performance comparison between Hermes and [1] in some scenarios, for example, FlagDynamic and CylinderFlow in [1].
>
> We did not initially compare to GNN approaches such as [1] since our primary focus is on determining when nonlinear message passing would be beneficial over simpler linear message passing schemes for gauge-equivariant networks and how decoupling the network depth from the receptive field of the neurons improves expressivity. The main difference between Hermes and MeshGraphNet is that Hermes learns directly on the two-dimensional mesh (mesh space) and does not depend on the embedding to 3D space (world space). It performs convolutions in a way that preserves local gauge symmetries and thus incorporates the geometry of the mesh intrinsically. On the other hand, MeshGraphNet uses both node positions in mesh space and world space as input node features and additionally creates edges between nodes that are close in world space (see Figure 3 of [1]). It also normalizes the input node and edge features and encodes them before performing several iterations of message passing.
>
> We agree that having more baselines to compare is useful. We have included MeshGraphNet [1] as a baseline for the PDE datasets. We scaled the hidden dimension and the number of message passing iterations so that the number of parameters is roughly equal to Hermes and keep the rest of the hyperparameters the same as in [1]. We do not use node/edge normalization to keep the RMSE values comparable. We also add two non-equivariant, non mesh-aware baselines (GCN and MPNN), a E(3)-equivariant non mesh-aware baseline (EGNN), and a non-equivariant, mesh-aware method (SpiralNet++). See Table 1 in the uploaded figures/tables page for the comparison of features of each method. Due to time constraints, we note that we were only able to conduct a coarse hyperparameter search for MeshGraphNet and mostly used the hyperparameters reported in the paper [4].
>
> Table 2 in the uploaded tables page shows that MeshGraphNet outperforms Hermes on Heat, does worse on Wave, and performs similarly or slightly worse on Cahn-Hilliard. Interestingly, we point out that MeshGraphNet does noticeably worse than Hermes on the test mesh dataset, suggesting that Hermes can more accurately learn the dynamics function and not just the specific to the seen trajectories. This property is important as one may use different discretizations and scales of the mesh when the underlying dynamics are the same.
> Since our method assumes triangular static meshes, we could not perform experiments on FlagDynamic and CylinderFlow in [1] and instead include FlagSimple. Table 3 shows that MeshGraphNet performs slightly better than Hermes.
>
> We will include these additional results in the final version.
>
> > In addition, Geo-FNO [2] can also perform PDE learning on irregular geometries, what is the advantages of Hermes over Geo-FNO in terms of accuracy in solving PDE on irregular domains.
>
> Thank you for the reference. The paper [2] indeed tackles similar problems. Geo-FNO assumes that there exists a diffeomorphic deformation between the embedding/input space and the computational mesh and then learns the deformation and then learns the Fourier neural operator in the computation space. A key difference between our method and Geo-FNO is that Hermes does not depend on the embedding space of the mesh (e.g. embedding a rough 2D mesh in 3D) and works directly on the intrinsic mesh surface. Geo-FNO mostly evaluates on flat 2D disks (with or without holes) where vertex normals would all be parallel and thus local symmetry would not be beneficial in these scenarios. We hypothesize that due to this difference, Hermes would outperform Geo-FNO on more irregular curvatures and rougher surfaces.  Furthermore, we note that the mesh sizes considered are < 2k vertices (we consider meshes with up to 4670 vertices). It would be interesting to test Hermes on different topologies such as the torus in future work. We will include this discussion in the final version.

---

> > ### Comment · Reviewer_ASko · 2023-08-17
> >
> > Thanks for taking the effort to conduct additional experiments. Most of my concerns have been addressed so I will increase the score to 5. And I highly recommend adding an experimental comparison with Geo-FNO in the camera-ready version.

---

> > > ### Author Response · Authors · 2023-08-20
> > >
> > > We thank the reviewer for raising their score and are glad that we have answered most of their concerns. We will definitely consider and look into adding Geo-FNO in the final version.

---

### Official Review · Reviewer_GgMe · 2023-07-07

**Soundness:** 4 excellent
**Presentation:** 4 excellent
**Contribution:** 3 good
**Rating:** 7
**Confidence:** 2

**Summary:**

This work proposes a novel graph neural network architecture for gauge-equivariant learning on meshes. Compared to prior work using convolutional or attentional methods, this paper shows a nonlinear message passing method for their gauge-equivariant graph neural network. A comparison with two baselines, GemCNN and EMAN, shows that the novel method, Hermes, outperforms baselines in a diverse set of example applications.

**Strengths:**

Great introduction to the problem setting, background well-explained step-by-step, making it very easy for the reader to follow. Example applications are clear, and always have arguments for why they are relevant problems in practice. Ablation studies also provide a full overview of the proposed method. A well-varied set of problems was used to show the performance increase of the proposed method.

**Weaknesses:**

1. In Line 143 the authors noted the addition of a residual connection to the HermesBlocks, for more expressivity. Was this perhaps tested in an ablation study? Did the model perform noticably better?

2. In Figure 3 the claim is that Hermes is accurate for all datasets, even though in the Wave example Hermes is diverging faster than EMAN.

3. Line 312 mentions there is a runtime increase and parameter increase involved with Hermes, this would be good to add to the main paper to understand the drawbacks of the method, as well as scalability. Although somewhat high-dimensional meshes (170k vertices) have already been tried out in this paper, it is unclear how much larger the authors are considering when mentioning the limitation in scalability in Line 314.

4. The paper concludes with an insightful remark, summarizing the findings from the paper. Ideally future work can be described as well, seeing what direction is most promising from the authors' opinion.



**Questions:**

1. For Table 1 the reported values of RMSE feel less interpretable than perhaps relative errors, since it might not be clear if all the datasets are normalized for the PDEs. Would relative errors make sense in this context?

2. What is the intuitive reason that all perform so poorly for the wave equation in rollout testing?

3. As far as understood, both GemCNN and EMAN in this paper have the gauge-equivariant adjustments, correct? Would it make sense to have a baseline that is not gauge-equivariant?

4. In Line 212 the authors mention how EMAN has far more parameters and constraining it would limit expressivity. Were the test examples hence using the constrained EMAN? I.e., did all models have the same number of parameters? It would also be interesting to see if the computation time during inference is different between the models, and how much they vary.

**Limitations:**

Very well detailed limitations are mentioned, and potential future work to tackle them.

---

> ### Author Rebuttal · Authors · 2023-08-10
>
> We thank the reviewer for the supportive and insightful feedback.
> > In Line 143 the authors noted the addition of a residual connection to the HermesBlocks ... tested in an ablation study?
>
> We performed an ablation study and results are shown in Table 5 of the uploaded page. The impact on model performance was mixed: having a residual connection improves performance on Heat, but decreases performance slightly on Wave and Cahn-Hilliard. We will include these results in the final version and make note that the residual connection should be considered as a task-dependent design dimension.
> > In Figure 3 the claim is that Hermes is accurate for all datasets, even though in the Wave example Hermes is diverging faster than EMAN.
>
> Thank you for pointing this out. We will adjust the wording to accurately reflect the results in the final version.
> > Line 312 mentions there is a runtime increase and parameter increase involved with Hermes ... mentioning the limitation in scalability in Line 314.
>
> We show the mean forward computation time during inference on the test time dataset in Table 4 of the uploaded page. Interestingly, Hermes is actually slightly faster than GemCNN, likely because Hermes performs fewer aggregations than GemCNN when using a similar number of parameters. Adding more message-passing iterations would increase the runtime over GemCNN.
>
> Regarding the number of vertices, we reduced the original meshes (~170k vertices) to <5000 vertices to keep training tractable. It is well known that GNNs do not scale to larger graphs [A, B] due to memory constraints. We are not aware of any message passing graph networks that can scale up to 170k vertices on a single GPU and scaling to this many vertices likely requires distributed computing and/or graph subsampling. Applying such techniques on Hermes to scale to larger graphs would be a good future research direction.
> > Ideally future work can be described as well, seeing what direction is most promising from the authors' opinion.
>
> Thank you for your comment. One future direction is to consider different dynamics such as non-stationary or chaotic dynamics and other PDEs important in real world applications. Another direction that is to analyze the design space of gauge-equivariant networks. While GNNs have been extensively studied, far less work exists for mesh methods. GNNs are often highly task-specific and there are many design dimensions (e.g. residual connections, message passing iterations, etc.) to consider [C]. It would be particularly helpful for practitioners to have guidelines on when to use gauge equivariance and/or message passing over simpler approaches. This work aims to be a first step in this direction by demonstrating Hermes as a good fit for predicting nonlinear dynamics on meshes.
> > For Table 1 the reported values of RMSE feel less interpretable than perhaps relative errors
>
> We chose RMSE as it would emphasize large differences from the ground truth and is a standard way to measure error [D,F]. While relative error is also a standard method and may be more interpretable as a unitless quantity, it also depends on where the zero point of the units is (e.g. measurements in Celsius vs. Kelvin would give different relative errors even though the absolute error is equivalent) and therefore relative error may not always be preferred over RMSE. We observe roughly 2-4% relative absolute errors for Hermes on the Heat and Cahn-Hilliard datasets.
>
> >What is the intuitive reason that all perform so poorly for the wave equation in rollout testing?
>
> One possible reason that all models diverge on long-term predictions is that the wave amplitude oscillates around 0 between [-1, 1] multiple times over the course of one training trajectory. On the other hand, the temperature in the heat PDE diffuses toward the mean and the concentration in the Cahn-Hilliard equation approach towards the boundaries. It is possible that it may be more difficult to predict the periodic nature of the wave and the exact times of the wave peaks.
> >Would it make sense to have a baseline that is not gauge-equivariant?
>
> Yes, both GemCNN and EMAN are gauge-equivariant methods. We considered this comparison most relevant as our primary consideration was to elucidate when nonlinear message passing is more beneficial than linear messages with respect to meshes. We have also added several new baselines: two standard GNN variants (GCN and MPNN), a non-equivariant mesh method (MeshGraphNet [D]), an E(3)-equivariant baseline (EGNN [E]), and a non-equivariant, mesh-aware method (SpiralNet++ [F]). See Table 1 for the comparison of features of each method and Table 2 for the results of the uploaded page. The results show that Hermes outperforms EGNN and SpiralNet++ on all datasets and also MeshGraphNet on test mesh.
> > ... did all models have the same number of parameters? ... computation time during inference.
>
> Yes, for our tests, we use a similar number of parameters for each method including EMAN (see Table 7 in the Appendix and Table 5 in the additional tables page). We also measured the mean computation time during inference on the  test time dataset and surprisingly find that Hermes is slightly better than GemCNN and roughly 1.5~2x than EMAN. All three gauge-equivariant methods are much slower than regular graph networks (e.g. GCN, EGNN, SpiralNet++). We will clarify the number of parameters and include the computation time in the final version.
>
> References:
> - [A] Wu, Z., et al. (2020). A comprehensive survey on graph neural networks.
> - [B] Duan, K., et al. (2022). A comprehensive study on large-scale graph training: Benchmarking and rethinking.
> - [C] You, J., et al. (2020). Design space for graph neural networks.
> - [D] Pfaff, T., et al. (2020). Learning Mesh-Based Simulation with Graph Networks.
> - [E] Satorras, V. G., et al. (2021). E (n) equivariant graph neural networks.
> - [F] Gong, S., et al. (2019). Spiralnet++: A fast and highly efficient mesh convolution operator.

---

> > ### Comment · Reviewer_GgMe · 2023-08-10
> >
> > Thank you for the extensive rebuttal and additional results. These very much strengthen the paper, and makes it a much more convincing novel framework. One questions that remains, is what the authors think of the new baseline with MeshGraphNet, and how this impacts the novelties of the proposed method. MeshGraphNet outperforms Hermes in some cases, and similarly in others, while running at much faster inference times. The authors mentioned in future work it would be interesting to see when to use gauge equivariance, how would the authors answer this question currently with the new results? Is there a clear area where gauge equivariance is beneficial whereas in other areas it is not? My other comments have been properly addressed, I'd like to thank the authors for their detailed response.

---

> > > ### Author Response · Authors · 2023-08-12
> > >
> > > We thank the reviewer for the quick and thoughtful reply and are glad that we have answered most concerns.
> > >
> > > It is true that in MeshGraphNet outperforms in Heat, but underperforms in Wave, and performs similarly to Hermes on Cahn-Hilliard. We would like to point that Hermes performs substantially better on the test mesh datasets, which may indicate that Hermes can generalize to the true dynamics function rather than the specific dynamics seen in the training trajectories.
> > >
> > > More generally in the equivariant networks literature, it is well known that using symmetry as an inductive bias guarantees generalization as opposed to data augmentation approaches, often leading to better sample efficiency and faster convergence. Gauge equivariance specifically improves generalization to local frame transformations and so Hermes would be likely be more useful in the small data regime, or where the local curvatures are not very homogeneous across the mesh. Although our experiments did not specifically evaluate sample efficiency, we can see some support for this as seen on the test mesh dataset, where Hermes outperforms MeshGraphNet on all PDE datasets.

---

### Author Rebuttal · Authors · 2023-08-10

## Summary of Response

We thank the reviewers for the detailed feedback and constructive suggestions and hope that we have addressed all concerns. The reviewers all appreciate the importance of the problem and the clear benefit of combining nonlinear message passing with a gauge-equivariant method for predicting complex dynamics. The main concern raised by reviewers seems to be more comparisons with standard baselines and on more datasets, and also an evaluation of computation time amongst the gauge equivariant methods.

We have added several new baselines with different features and also a new dataset for a stronger evaluation. We display the main results here and have also added all results in the additional figures/tables page.

Summary of additions:

1. Added several new baselines: non-equivariant GNN variants (GCN and MPNN), a SOTA message passing, mesh-aware method  (MeshGraphNet), an equivariant, non mesh-aware message passing network (EGNN), a non-equivariant, mesh-aware baseline (SpiralNet++). Results are shown in Table 2 of the uploaded figures/tables page.
2. Included a detailed comparison of features of each method in Table 1 of the uploaded figures/tables page.
3. Added a new FlagSimple dataset to compare Hermes and MeshGraphNet (Table 2 of uploaded page).
4. Recorded the average computation time during inference for each method in Table 3 and discussed the difference in computation between GemCNN, Hermes, and non gauge-equivariant methods.
5. Performed an ablation study on residual connections in Hermes.

Additional results with new baselines and new dataset:

|  |   | Hermes | GCN | MPNN | MeshGraphNet | EGNN | SpiralNet++
|---|---|---|---|---|---|---|---|
|Heat  | Test time ($\times 10^{-3}$) | $1.18 \scriptstyle \pm 0.3$ | $152 \scriptstyle \pm 1.2$ | $2.66 \scriptstyle \pm 0.8$|$\textbf{0.93} \scriptstyle \pm 0.2$ | $3.09 \scriptstyle \pm 1.2$ | $2.82 \scriptstyle \pm 0.2$
|   | Test init ($\times 10^{-3}$) | $1.16 \scriptstyle \pm 0.3$ | $152 \scriptstyle \pm 0.9$ | $2.63 \scriptstyle \pm 0.8$|$\textbf{0.93} \scriptstyle \pm 0.2$ | $3.07 \scriptstyle \pm 1.2$ | $6.44 \scriptstyle \pm 0.1$
|   | Test mesh ($\times 10^{-3}$) |$\textbf{1.01} \scriptstyle \pm 0.3$ | $127 \scriptstyle \pm 2.2$ | $2.36 \scriptstyle \pm 0.7$|$2.41 \scriptstyle \pm 1.1$ | $8.96 \scriptstyle \pm 5.0$ | $22.0 \scriptstyle \pm 0.2$
|Wave  | Test time ($\times 10^{-3}$) | $\textbf{5.43} \scriptstyle \pm 0.8$ | $162 \scriptstyle \pm 5.0$ | $9.07 \scriptstyle \pm 1.2$|$6.26 \scriptstyle \pm 0.9$ | $45.9 \scriptstyle \pm 6.1$ | $8.88 \scriptstyle \pm 1.2$
|   | Test init ($\times 10^{-3}$) | $\textbf{3.72} \scriptstyle \pm 1.3$ | $158 \scriptstyle \pm 5.9$ | $5.24 \scriptstyle \pm 1.1$|$4.24 \scriptstyle \pm 0.6$ | $12.1 \scriptstyle \pm 3.5$ | $8.47 \scriptstyle \pm 0.6$
|   | Test mesh ($\times 10^{-3}$) | $\textbf{3.79} \scriptstyle \pm 1.3$ | $164 \scriptstyle \pm 5.1$ | $6.29 \scriptstyle \pm 1.3$ | $7.01 \scriptstyle \pm 1.9$ | $54.5 \scriptstyle \pm 18$ | $10.8 \scriptstyle \pm 0.8$
|Cahn-Hilliard  | Test time ($\times 10^{-3}$) | $\textbf{4.23} \scriptstyle \pm 0.9$ | $250 \scriptstyle \pm 7.6$ | $7.25 \scriptstyle \pm 3.1$|$\textbf{4.49} \scriptstyle \pm 0.6$ | $8.36 \scriptstyle \pm 1.4$ | $11.6 \scriptstyle \pm 3.3$
|   | Test init ($\times 10^{-3}$) | $5.21 \scriptstyle \pm 1.2$ | $383 \scriptstyle \pm 6.0$ | $7.52 \scriptstyle \pm 3.0$|$\textbf{4.64} \scriptstyle \pm 0.6$ | $10.6 \scriptstyle \pm 1.1$ | $12.9 \scriptstyle \pm 2.8$
|   | Test mesh ($\times 10^{-3}$) | $\textbf{5.34} \scriptstyle \pm 0.9$ | $391 \scriptstyle \pm 8.6$ | $7.63 \scriptstyle \pm 3.0$|$18.7 \scriptstyle \pm 7.2$ | $9.38 \scriptstyle \pm 1.7$ | $13.4 \scriptstyle \pm 2.5$
|Flag  | Test ($\times 10^{-3}$) | $\textbf{5.87} \scriptstyle \pm 0.0$| - | - |$\textbf{9.01} \scriptstyle \pm 0.1$ | - | - |

Average forward computation time during inference of each considered method on the test time dataset.

|  | GemCNN | EMAN  | Hermes | GCN | MPNN | MeshGraphNet | EGNN | SpiralNet++
|---|---|---|---|---|---|---|---|---|
|Heat (s) | 0.0124 | 0.0195 | 0.0108 | 0.0015 | 0.0012 | 0.0022 | 0.0014 | 0.0009 |
|Wave (s) | 0.0125 | 0.0177 | 0.0105 | 0.0014 | 0.0013 | 0.0022 | 0.0013 | 0.0009 |
|Cahn-Hilliard (s) | 0.0069 | 0.0092 | 0.0062 | 0.0019 | 0.0015 | 0.0021 | 0.0016 | 0.0013 |

---

### Decision · Program_Chairs · 2023-09-21

**Decision:**

Accept (poster)

**Comment:**

All reviewers but one are strongly in favor of this work, and it appears to have both mathematical and practical value. The AC agrees with the positive assessments. In the final revision, please incorporate promised changes from the rebuttal, and importantly please include the new experiments introduced in the rebuttal.